# 3D hierarchical graphene matrices enable stable Zn anodes for aqueous Zn batteries

Yongbiao Mu[1,2,3,5], Zheng Li[1,2,3,5], Bu-ke Wu[1,2,3,5], Haodong Huang[1,2,3], Fuhai Wu[1,2,3], Youqi Chu[2], Lingfeng Zou[1,2,3], Ming Yang[2,4], Jiafeng He[2,3], Ling Ye[2], Meisheng Han[2,3], Tianshou Zhao[1,2,3] ✉ & Lin Zeng [1,2,3] ✉

Metallic zinc anodes of aqueous zinc ion batteries suffer from severe dendrite and side reaction issues, resulting in poor cycling stability, especially at high rates and capacities. Herein, we develop two three-dimensional hierarchical graphene matrices consisting of nitrogen-doped graphene nanofibers clusters anchored on vertical graphene arrays of modified multichannel carbon. The graphene matrix with radial direction carbon channels possesses high surface area and porosity, which effectively minimizes the surface local current density, manipulates the $Zn^{2+}$ ions concentration gradient, and homogenizes the electric field distribution to regulate Zn deposition. As a result, the engineered matrices achieve a superior coulombic efficiency of 99.67% over 3000 cycles at 120 mA cm$^{-2}$, the symmetric cells with the composite zinc anode demonstrates 2600 h dendrite-free cycles at 80 mA cm$^{-2}$ and 80 mAh cm$^{-2}$. The as-designed full cell exhibits an inspiring capacity of 16.91 mAh cm$^{-2}$. The Zn capacitor matched with activated carbon shows a superior long-term cycle performance of 20000 cycles at 40 mA cm$^{-2}$. This strategy of constructing a 3D hierarchical structure for Zn anodes may open up a new avenue for metal anodes operating under high rates and capacities.

Metallic zinc (Zn) has garnered significant attention as an anode material for aqueous zinc-ion batteries (AZIBs) due to its high theoretical capacity (820 mAh g$^{-1}$, 5855 mAh cm$^{-3}$), low redox potential (−0.76 V vs. SHE), and environmentally friendly and safe properties[1–3]. These advantages position AZIBs as one of the most promising energy storage technologies for practical applications. However, challenges such as Zn dendrite growth, side reactions like hydrogen evolution reaction (HER), and Zn corrosion significantly impact the reversibility and cycling performance of these batteries, limiting their widespread commercial use[4–6]. The formation of hazardous zinc dendrites is particularly problematic at high areal capacities and current densities, leading to non-uniform Zn plating/stripping, extensive dendritic growth, and volumetric expansion. These issues further accelerate

hydrogen evolution and Zn corrosion due to increased Zn exposure areas[7]. Moreover, at high current densities and capacities, rapid charge-discharge processes exacerbate the problem, resulting in thick Zn plating and incomplete zinc stripping.

Several strategies have been proposed to address those issues initiated by Zn anodes, such as surface modification (MOF-based materials[8,9], In[10], carbon spheres[11], three-dimensional (3D) porous ZnO[12], TiO$_2$/PVDF[13,14]), electrolyte optimization (electrolyte additives[15–17], gel electrolytes[18], and flowing electrolyte[19]), separator selection (polypropylene membranes[20], polysulfonium films[21], polybenzimidazole (PBI) membrane[22], Nafion 211 membrane[23], and PAN polymer films[24]) and electrode structural design (Zn foil[25], Zn powder[26,27], 3D porous architectures[28–30]). Firstly, various modified

[1]Shenzhen Key Laboratory of Advanced Energy Storage, Southern University of Science and Technology, Shenzhen 518055, China. [2]Department of Mechanical and Energy Engineering, Southern University of Science and Technology, Shenzhen 518055, China. [3]SUSTech Energy Institute for Carbon Neutrality, Southern University of Science and Technology, Shenzhen 518055, China. [4]College of Chemistry and Environmental Engineering, Shenzhen University, Shenzhen 518060, China. [5]These authors contributed equally: Yongbiao Mu, Zheng Li, Bu-ke Wu. ✉e-mail: zhaots@sustech.edu.cn; zengl3@sustech.edu.cn

surface coating on the Zn anodes have been commonly used to isolate the electrode and electrolyte, inhibit Zn dendrites and mitigate Zn corrosion by providing a homogeneous electric field, ion flux, and physical barriers[31,32]. Advanced functional coatings mainly include metals, metal oxides, bimetallic hydroxides, carbon materials, metal-organic frameworks, and non-conductive materials[33–35]. Secondly, the composition, concentration, and format of the electrolyte play a crucial role in the reversible capacity and rechargeability of the battery. To optimize the electrochemical performance of Zn and address challenges such as Zn corrosion, passivation, shape change, and dendrite growth, various electrolyte additives, and gel electrolytes have been proposed. These strategies aim to suppress Zn corrosion, mitigate Zn passivation, minimize shape change, and inhibit the growth of Zn dendrites[36–38]. Besides, a series of functional electrolytes were applied to optimize the ion transfer rate as well as the ion flux, regulate the growth of Zn crystal and contribute to high rates/capacities and prolonged lifespan[39,40]. Furthermore, the 3D conductive matrix was employed as a current collector to minimize polarization by reducing the local current density and alleviating the Zn dendrite formation[41,42]. Among these methods, both dendrite growth and cycle lifespan of Zn anodes were mitigated to some extent, however, there are still some unsolved issues with these strategies, which restrict them to subdued performance levels in Zn batteries. For example, the huge volume change during repeated Zn plating/stripping could damage the interfacial modification layers and even peel them off the Zn matrix. Meanwhile, most of the previous strategies were tested and operated based on relatively mild conditions, which is difficult to meet the security of extreme working conditions. Therefore, Zn dendrite at high current ($40\,mA\,cm^{-2}$) and high capacity ($40\,mAh\,cm^{-2}$) remains a serious issue, which leads to poor electrochemical performance and low operability under extreme conditions. As a result, it is of great importance to take a different design and approach to alleviate Zn dendrites and enable the operation of Zn batteries with high current densities and capacities.

3D conductive matrices, serving as current collectors, are crucial for battery performance and cost optimization[43,44]. Designing a 3D porous architecture is a widely adopted approach to address the challenges associated with Zn electrodes. This is attributed to the large specific surface area, which reduces the local current density on the 3D Zn electrode, resulting in low overpotential and a slower Zn deposition process. Carbonaceous materials, known for their high conductivity, lightweight nature, and ease of fabrication, are commonly used as conductive matrices for Zn anodes. Various carbon-based materials, including carbon fibers[45], graphene foams[25], graphene oxide[46], carbon nanotube frameworks[47], defective carbon[48], and biomass-derived carbon[49], have been employed. Porous metal frameworks such as Cu/Ni foam/mesh, exhibiting high electrical conductivity, porosity, and improved zincophilicity, also show promise as matrices for Zn anodes. However, the design of 3D substrates for AZIBs still faces several challenges. For instance, dendritic Zn deposition is still observed on these substrates, likely due to limited nucleation sites on the relatively smooth skeleton surface. Additionally, the binding force between the deposited Zn and the matrix is weak, resulting in numerous cracks or defects. Moreover, Zn deposits on the upper surface of the 3D architectures, particularly at high current densities and capacities, leading to wasted 3D spaces. Therefore, there is an urgent need to develop alternative 3D matrices that enable dendrite-free Zn anodes while maintaining homogeneous Zn plating/stripping at high rates and capacities.

In this contribution, we demonstrate a novel 3D composite Zn anodes with ultrahigh Zn plating/stripping rates and capacities as well as superior long-term cycling life, which is constructed of 3D nitrogen-doped graphene nanofibers clusters (GFs) anchored in vertical graphene arrays (VGs) modified multichannel carbon matrix (3D-FGC). The thermal-chemical vapor deposition (T-CVD) strategy realizes the

growth of GFs and VGs over the original 3D carbon matrices. The Zn metal was deposited along with the longitudinal direction (3D-LC; 3D-LFGC) and radial direction (3D-RC; 3D-RFGC) of the 3D multichannel carbon matrices to investigate the plating and stripping behaviors, compared with bare Zn foil and Cu foil substrates, as shown in Fig. 1. As a result, the 3D-RFGC@Zn anodes show prolonged cycle life and improved reversibility in large current densities ($40–120\,mA\,cm^{-2}$) and surface areal capacities ($1–80\,mAh\,cm^{-2}$). By virtue of the 3D hierarchical-structure graphene matrices, at $80\,mA\,cm^{-2}$, an AZIBs anode can work smoothly and steadily with an ultrahigh areal capacity of $80\,mAh\,cm^{-2}$ for ~2600 h with low overpotential (<55 mV) in a symmetric cell. The 3D-RFGC@Zn composite anode delivers superb coulombic efficiency (CE) values of 97–99% at current densities of $1–120\,mA\,cm^{-2}$ in half cells. When paired with 3D $V_2O_5$ ($V_2O_5$@3D-LC) and 3D $MnO_2$ ($MnO_2$@3D-LC) cathodes, the full cells deliver remarkably high rates and cycling stability (2500 cycles at $40\,mA\,cm^{-2}$ for $V_2O_5$ and 500 cycles at $4\,mA\,cm^{-2}$ for $MnO_2$). More encouragingly, the AC@3D-LC/3D-RFGC@Zn capacitor shows a superior long-term cycle performance of 20000 cycles at $40\,mA\,cm^{-2}$.

## Results
### Synthesis and characterization of 3D-FGC matrices
Compared with the planar Zn foil, the 3D hierarchical graphene structure facilitates the uniform deposition of Zn metal and inhibits the growth of Zn dendrites. The hypothesis of the deposition behaviors of Zn metal on the bare Zn and 3D graphene matrices are illustrated in Fig. 1. For the bare Zn foil or 2D Cu substrate, Zn metal nucleates at high $Zn^{2+}$ concentrations. The Zn dendrite grows rapidly and ultimately penetrates the separator during repeated cycles, making the cells short-circuit and causing safety hazards, especially at high rates and capacities. On the contrary, the deposition of Zn metal on the 3D matrices is different. The Zn metal deposits on the abundant graphene skeletons due to its zincophility, gradually fill the pores to form dense 3D Zn anodes. What is more, for both 3D-LFGC (Fig. 1b) and 3D-RFGC (Fig. 1c) graphene matrices, there is enough space to store Zn because of their numerous pores on the surface, which realizes high utiliazation of graphene modification layer, thus obtaining high areal capacity. Even at high current densities, the plating and stripping of Zn metal cannot be out of control and result in dendrite growth, which ascribes low local current distribution to their large specific surface area and abundant deposition sites. In addition, the continuous porous structure can also buffer the volume change of deposited Zn during the deposition and dissolution within the 3D-RFGC matrix.

The hierarchical-structure graphene harnessing 3D N-doped graphene nanofibers clusters (GFs) and vertical graphene arrays (VGs) modified multichannel carbon matrices (3D-LFGC/3DD-RFGC) was developed via a sequential two-step approach, as shown in Fig. 2a. The pristine carbon channels began with natural woodblocks from the longitudinal direction (Fig. S1, Figure S2) and radial direction (Fig. S3). In the first step, biomass wood was converted into multichannel carbon matrices through stabilization and carbonization processes (Fig. S4), as shown in the top views (3D-LC) (Fig. S5a, b) and cross-sectional views (3D-RC) (Fig. S5d, e). The micro CT images further demonstrate the porous and continuous structure (Fig. S5c, f), which is beneficial to deposit Zn metal and loading cathode materials of ZIBs. Subsequently, the GFs and VGs were grown on the 3D-LD and 3D-LC simultaneously by a thermal-chemical vapor deposition (T-CVD) method without any catalyst. The mechanism of growing the GFs and VGs was discussed with and without metal catalytes on different substrates (for instance, carbon particles, carbon nanofibers, carbon fibers) (Fig. S6). The growth of GFs ascribes to the existence of trace elements (Mg, Ca, Fe, Zn, Na) in pristine carbon mtrices (Fig. S7), which was demonstrated by introducing ferric nitrate ($Fe(NO_3)_3 \cdot 9H_2O$) as catalytic source (Fig. S8). The T-CVD strategy provides a simple and rapid strategy to prepare large-area vertical graphene materials. A

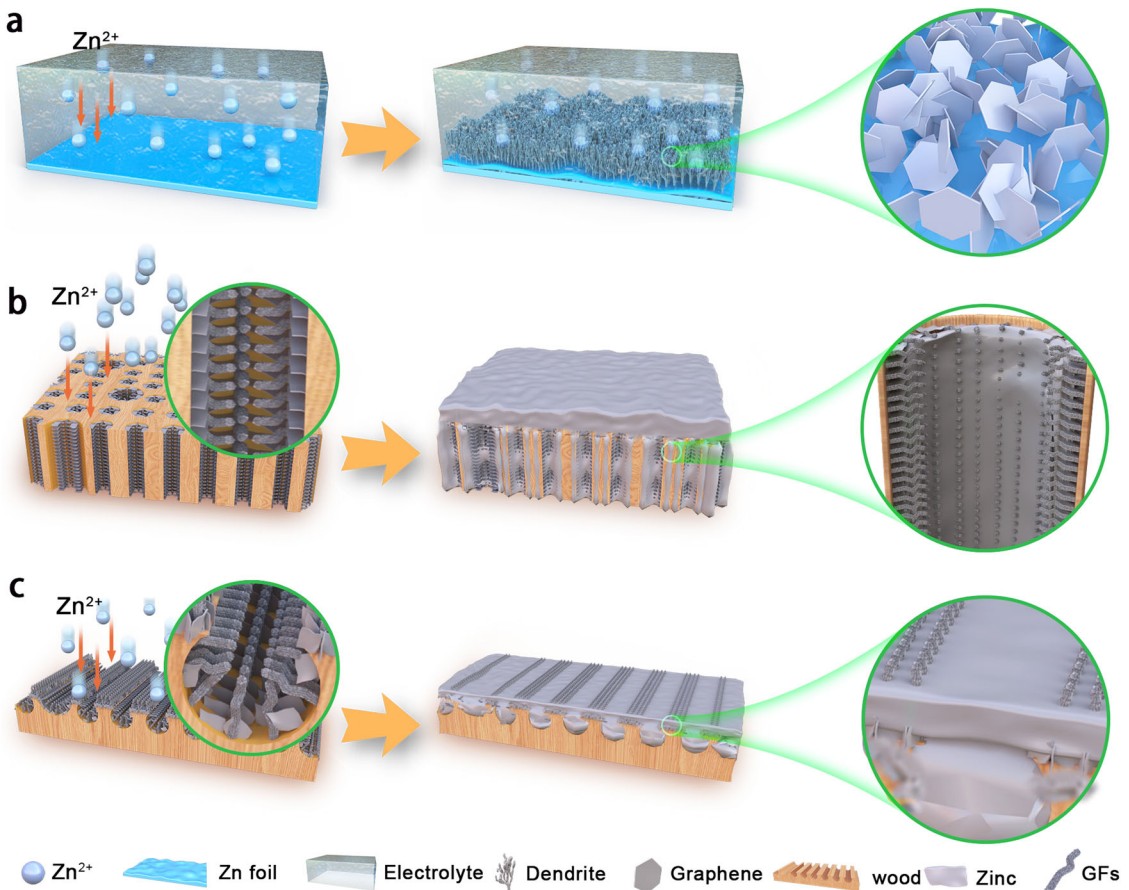

**Fig. 1 | Schematic illustrating the design of 3D hierarchical-structure graphene matrices targeting stable Zn anodes.** Zn deposition on **a** bare Zn, **b** 3D graphene matrices of longitudinal direction (3D-LFGC), and **c** radial direction (3D-RFGC).

$20 \times 10$ cm$^2$ 3D-RFGC matrices was obtained only within 2 h (Fig. S3). Scanning electron microscopy (SEM) and transmission electron microscopy (TEM) observations reveal that the GFs and VGs can form a hierarchical structure with large space and highly efficient conductive networks (Fig. 2b). The morphology of 3D-LFGC reveals the channel size ranges from 10 to 30 μm, typically at 20 μm. Specifically, GFs with a diameter of 250 nm and VGs were firmly anchored to the interior of channels, providing a 3D interconnected and versatile framework (Fig. 2c–e). The length of GFs and the height of VGs are controllable by varying the growth time from 2 h to 12 h for both the longitudinal direction (Fig. S9) and radial direction (Fig. 2f, g, S10). TEM images confirm that the VGs were successfully grown on both carbon channels (Fig. 2h, i) and GFs (Fig. 2j). The selected area electron diffraction (SAED) pattern measured from the 3D-RFGC matrices comprises four concentric rings, well corresponding to the (002), (100), (102), and (110) planes of the graphitic structure, respectively. A high-resolution TEM image points out few layers of graphene structure with an interlayer spacing of 0.34 nm, consistent with that of the previous reports (Fig. 2k)[50]. TEM mapping displayed the successful doped of nitrogen and oxygen in the 1D and 2D carbon structure, which is helpful in creating zincophilic surfaces (Fig. 2l). X-ray diffraction (XRD) patterns of 3D-LFGC and 3D-RFGC show higher intensity and upshifted peaks, corresponding to the (002) peak (Fig. 2m). Raman signals of graphene at 1340 cm$^{-1}$ (*D* band), 1570 cm$^{-1}$ (*G* band), and 2660 cm$^{-1}$ (*2D* band), are indicative of the successful formation of graphene over the carbon channels and GFs by the direct T-CVD route. The intensity ratio (-2.05) of *G* band to *D* band ($I_G/I_D$) for the 3D-FGC is higher than that (-0.89) for the original carbonized wood, which is primarily related to the defects or turbostratic carbons of GFs and VGs within the carbon channels

(Fig. 2n). Based upon the $I_G/I_D$, Raman mapping (Fig. 2o) further indicates the uniformity of GFs and VGs coatings in hierarchical-structure design, consistent with the SEM and TEM observations. The X-ray photoelectron spectroscopy (XPS) survey result of the high-resolution N *1 s* spectrum confirms the presence of nitrogen-doping (Fig. 2p). The nitrogen adsorption isotherm of the as-prepared samples indicates a negligible difference between the pristine carbon and 3D graphene matrices (Fig, S10). A specific surface area (SSA) of 181.06 m$^2$g$^{-1}$ for 3D-RFGC was achieved, which enable Zn$^{2+}$ ions to access the abundant sites during the electrochemical process (Fig. 2q). To probe the wettability between 3D graphene matrices and electrolytes (2 M ZnSO$_4$), the electrolyte wetting processes of Zn foil, 3D-LC, 3D-RC, 3D-LFGC and 3D-RFGC were also recorded (Fig. S11). Improved electrolyte wettability demonstrates the uniform nitrogen-doping and abundant active sites on 3D graphene matrices (Fig. 2r, Table S1). The configurations of zinc adsorption on different sites on carbon substrates are depicted in Figs. S13 and S14. The binding energies of zinc atoms on pristine graphene at various adsorption sites are −0.0196 eV (Top), −0.0203 eV (Hollow), and −0.0180 eV (Bridge), respectively. In contrast, on N-doped graphene, the corresponding binding energies are −0.0246 eV (Top), −0.0270 eV (Hollow), and −0.0254 eV (Bridge) (Fig. 2s). These results highlight that the introduction of nitrogen-doping alters the interaction between zinc and the carbon substrate from zincophobic to zincophilic.

## Evolutions of Zn deposition/dissolution efficiency under different conditions
The evolutions of Zn deposition/dissolution efficiency morphologies on Cu foil, Zn foil, 3D-LC, 3D-RC, 3D-LFGC, and 3D-RFGC substrates

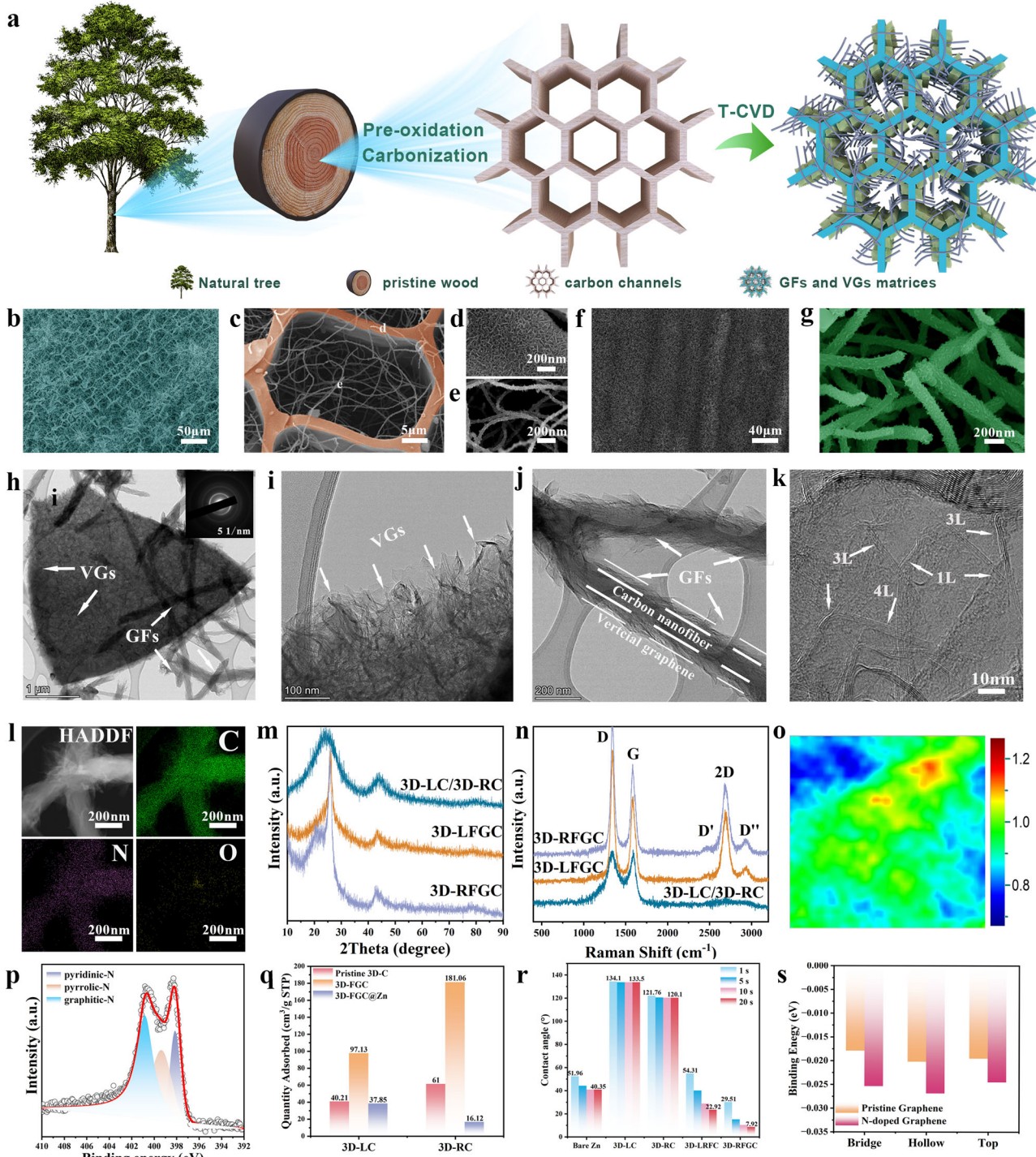

**Fig. 2 | Structural and morphological characterizations. a** Schematic illustration of fabricating the 3D-LFGC and 3D-RFGC matrices; SEM images of **b**, **c** 3D-LFGC matrix, **d** VGs, **e** GFs, **f** 3D-RFGC matrices, **g** GFs image. **h** TEM images of the GFs and VGs grown on the 3D carbon channels (Inset is SAED image, **i** VGs, **j** GFs, **k** high-resolution TEM of graphene nanosheets, and **l** the corresponding elemental mappings of individual GFs. **m** XRD patterns, **n** Raman spectrum, and **o** Raman mapping of 3D-RFGC matrices. **p** High-resolution of N *1 s* elements. **q** BET data of various carbon matrices; **r** contact angle at different times within 20 s and **s** binding energy of pristine graphene and N-doped graphene on different sites.

during galvanostatic charge-discharge processes were probed at varying current densities (1-120 mA cm⁻²) with capacities from 0.5 mAh cm⁻² to 10 mAh cm⁻². As shown in Fig. 3a, the overpotentials of the 3D-LFGC and 3D-RFGC electrodes are 32 and 30 mV at 1.0 mA cm⁻², respectively. In contrast, the Cu foil electrode exhibits quite a sharp voltage dip at the nucleation stage, showing a large overpotential of 52 mV. Ex-situ SEM images reveal that the Cu foil was gradually covered

by deposited Zn metal with the deposition capacities increase (Fig. S15), which leads to distinct lamellar porous dendrites. This result can be ascribed to the inhomogeneous plating of Zn on the 2D substrates, which is similar to the Zn foil (Fig. S16). Furthermore, the 3D-LFGC and 3D-RFGC matrices exhibit a more stable overpotential even at high current densities of 2, 5, and 10 mA cm⁻² (Fig. 3b–d). A homogeneous and dense surface morphology could be observed for

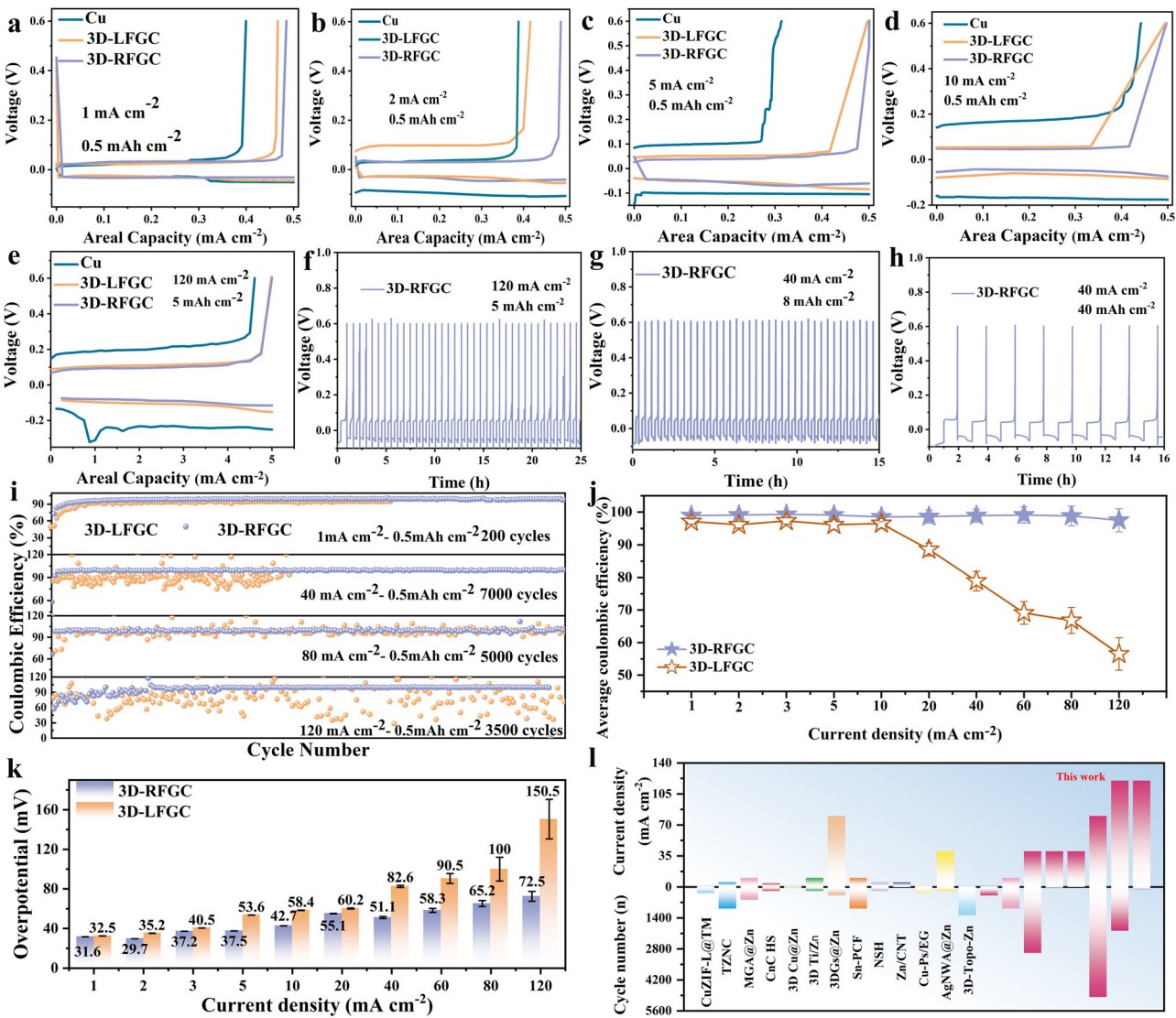

**Fig. 3 | Electrochemical performance of different electrodes in Zn/3D-LFGC, Zn/3D-RFGC, and Zn/Cu half cells.** Voltage profiles of different half cells at **a** 1 mA cm⁻², **b** 2 mA cm⁻², **c** 5 mA cm⁻², and **d** 10 mA cm⁻² with a capacity of 0.5 mAh cm⁻²; **e, f** 120 mA cm⁻² with capacity of 5 mAh cm⁻²; **g** 40 mA cm⁻² with capacity of 8 mAh cm⁻²; **h** 40 mA cm⁻² with capacity of 40 mAh cm⁻². **i** Zn plating/ stripping CE in different electrodes at varied current densities, **j** the corresponding comparison of CE; **k** the overpotential of Zn deposition on 3D-LFGC and 3D-RFGC at different current density; **l** the comparison of current densities of deposition, and cycle capability with previously reported literature.

3D-LFGC and 3D-RFGC matrices due to the low local current density and uniform $Zn^{2+}$ ions distribution originating from the abundant porous structure (Fig. S17). At a current density of up to 120 mA cm⁻², excellent deposition/dissolution overpotential and efficiency can still be realized for 3D-RFGC with a capacity of 40 mAh cm⁻² (Fig. 3e, f). When the plating capacity increased from 1 to 40 mAh cm⁻², most of the voids between the GFs and VGs were filled and the porosity of the electrode decreased, indicating that the Zn deposition occurred in the spaces of 3D-LFGC (Fig. S18a–d) and 3D-RFGC matrices (Fig. 3g, S19a–d). The curves of Zn palting/stripping on 3D-RFGC show a low overpotential of 76 mV at a current density of 40 mA cm⁻² and capacity of 40 mAh cm⁻² (Fig. 3h). However, more Zn dendrites were observed for bare Zn anode when plating capacity increases (Fig. S20a–d). As shown in Fig. 3i, a stable Zn plating/stripping behavior on 3D-RFGC with CE values between 97.5% and 99.52% at the current densities from 1 to 120 mA cm⁻² (Fig. S21). The obvious advantages of 3D-RFGC over 3D-LFGC in the deposition/dissolution efficiency can be found at high current densities beyond 10 mA cm⁻² (Fig. 3j). The 3D-RFGC matrices show a low plateau overpotential of the 5th cycle at various current

densities compared with the 3D-LFGC matrices, which demonstrate low local current density and homogeneous $Zn^{2+}$ ions distribution (Fig. 3k). This is ascribed to the good wettability between 3D matrices and electrolyte and strong interaction of zinc and N-doped VGs and GFs skeletons that promotes homogeneous $Zn^{2+}$ flux distribution and dense Zn deposition. Without GFs and VGs, it was clear that the Zn was merely deposited on the upper of the pristine 3D-LC (Fig. S22a–d), which results in poor spatial utilization of the 3D matrices. Rapid dendrite growth could occur as the increasing Zn deposition thickness, especially at high rates and capacities. Similarly, it is obvious that overpotential values start to increase for 3D-LFGC matrices when the current densities exceed 10 mA cm⁻² due to the incomplete plating/ stripping of Zn in the vertically-aligned channels (Fig. 3k).

Therefore, these findings provide strong evidence that the surface 3D hierarchical graphene structure effectively enhances the reversibility of Zn plating/stripping, thereby stabilizing Zn anodes through the reduction of local current density, accommodation of volume change, and homogenization of interfacial charge distribution. The impressive combination of high Coulombic efficiency (CE) values and

extended lifespan achieved by the 3D hierarchical graphene matrices is comparable to, and in some cases even surpasses, those reported for previous 3D Zn anodes (Fig. 3l, Table S2).

## Performances of 3D Zn anodes in symmetrical cells

Symmetric cells were assembled to determine the long-term cycling performance of 3D-LFGC@Zn, 3D-RFGC@Zn, 3D Zn foam (Fig. S23), and bare Zn foil according to galvanostatic charge and discharge (GCD) at current densities from 1 to 80 mA cm$^{-2}$ and areal capacities

from 1 to 80 mAh cm$^{-2}$, as shown in Fig. 4. It is evident that 3D-RFGC@Zn and 3D-LFGC@Zn anodes show reduced voltage hysteresis (21 mV) and significantly prolonged lifetime (7300 h for 3D-RFGC@Zn and 4200 h for 3D-LFGC@Zn) compared to bare Zn and 3D Zn foam (Fig. 4a, b, S24a). The enlarged voltage curves indicate the details of Zn plating/striping on different substrates (Fig. 4c–e). The lifespan of 3D-RFGC@Zn anode is extended in a wide range of applied current density and capacity, including 820 h for 2 mA cm$^{-2}$/2 mAh cm$^{-2}$, 400 h for 5 mA cm$^{-2}$/5 mAh cm$^{-2}$, 400 h for 8 mA cm$^{-2}$/8 mAh cm$^{-2}$, 200 h for

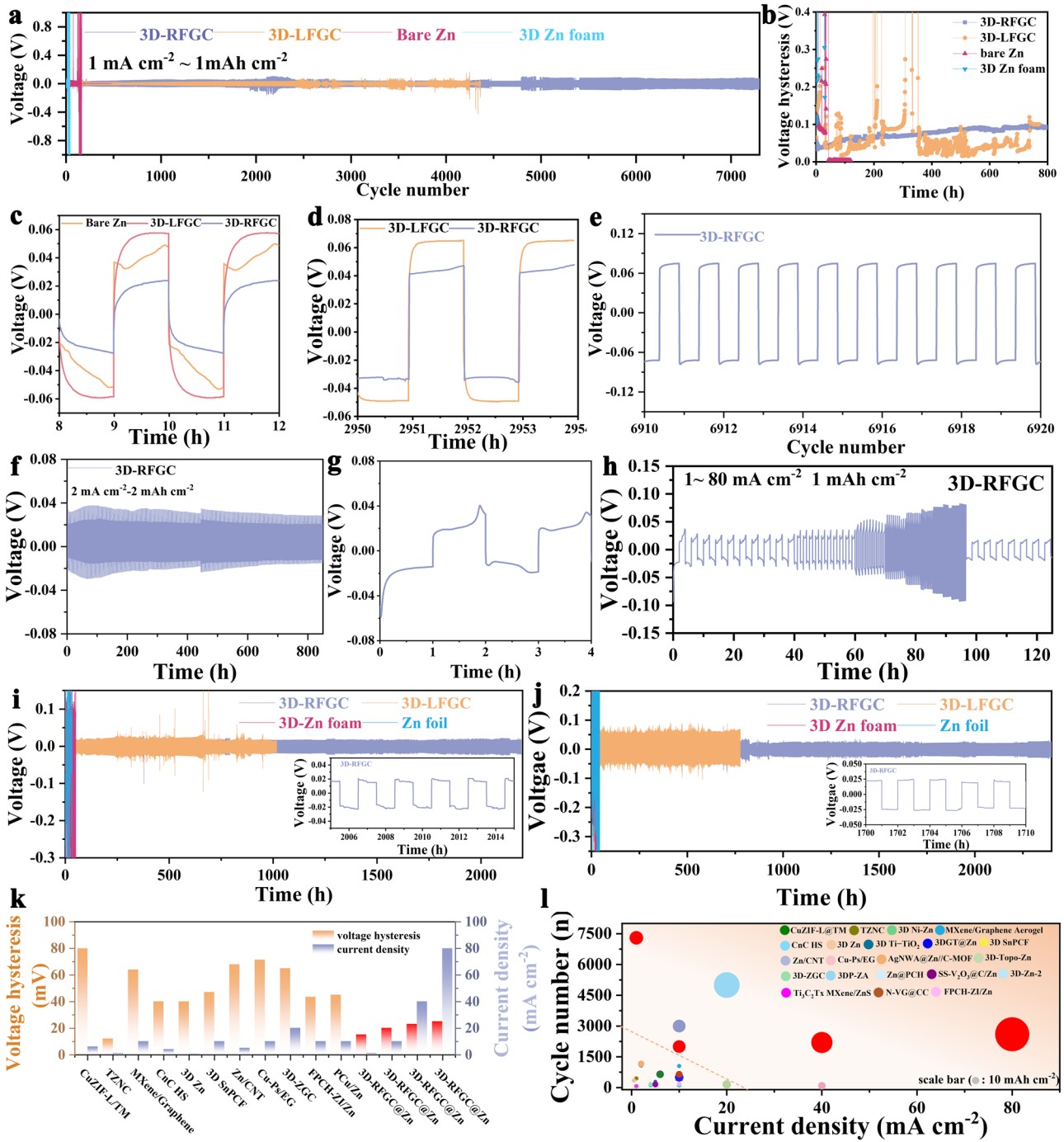

**Fig. 4 | Electrochemical performance of various Zn anodes in symmetrical cells.** **a** The symmetrical cells of bare Zn, 3D Zn foam, 3D-LFGC@Zn, and 3D-RFGC@Zn anodes at 1 mA cm$^{-2}$ and 1 mAh cm$^{-2}$; **b** the corresponding voltage hysteresis curve; **c–e** magnified voltage profiles for different electrodes at different stages. Symmetrical cells operating at **f** 2 mA cm$^{-2}$ and 2 mAh cm$^{-2}$; **i** 40 mA cm$^{-2}$ and

**j** 80 mA cm$^{-2}$ and 80 mAh cm$^{-2}$. **g** The enlarged voltage curves of Fig. 4f; **h** rate performance of 3D-RFGC@Zn anode. The comparison of **k** voltage hysteresis and **l** current densities, areal capacities, and cycle capability in symmetrical cells with previously reported 3D Zn anodes literature (the size of the circles represents the values of areal capacities).

20 mA cm$^{-2}$/20 mAh cm$^{-2}$, respectively) (Fig. 4f, g and S25). Moreover, the rate capability of 3D-RFGC@Zn anode in symmetric cells is also probed (Fig. 4h). The 3D-RFGC@Zn anodes maintain a smaller voltage hysteresis at the different current densities from 1.0 to 80.0 mA cm$^{-2}$ compared with the bare Zn-based symmetric cells (Fig. S26). Surprisingly, upon the harsh operating conditions, the 3D-RFGC@Zn anode also exhibits excellent cycling stability. Typically, the 3D-RFGC@Zn symmetrical cell can still maintain stable cycling for more than 2200 h, whereas the bare Zn and 3D Zn foam anodes display fluctuating curves before a short circuit only around 20 h under 40 mA cm$^{-2}$/40 mAh cm$^{-2}$ and 80 mA cm$^{-2}$/80 mAh cm$^{-2}$ (Fig. 4I, j, S24b–d). The 3D-RFGC@Zn anode shows substantially small voltage hysteresis, especially at high rates. Meanwhile, the morphologies of the 3D-RFGC@Zn at various current densities were investigated by SEM. 3D-RFGC@Zn anodes present a smooth surface with dense morphology. With the rates increase, the size of deposited Zn metal enlarged gradually in the form of planar stacking. No Zn dendrites can be observed (Fig. S27). The highly competitive voltage hysteresis and the ultrahigh current density as well as the areal capacity of symmetric cells with 3D-RFGC@Zn were compared with those of the previously reported 3D Zn hosts (Fig. 4k, l), indicating that our newly designed surface 3D hierarchical graphene matrices with abundant N-doped GFs and VGs as well as large volume channels effectively suppress the dendrite growth, leading to a highly reversible Zn plating/stripping process.

**Exploration of Zn deposition mechanism on the 3D graphene matrices**

In situ observations were conducted during the Zn metal plating process on both bare Zn foil and 3D-RFGC matrices using an optical microscope at a current density of 40 mA cm$^{-2}$. On the bare Zn foil, after 10 min, small and irregularly-shaped Zn particles were observed on the surface, accompanied by uneven Zn deposition and the formation of bubbles, indicating possible hydrogen production. After 20 min, visible protuberances appeared, and moss-like Zn dendrites gradually formed due to the uneven deposition (Fig. 5a). In contrast, the 3D-RFGC matrices showed minimal changes with a smooth surface throughout the plating process, maintaining a dendrite-free morphology even after 60 min of deposition (Fig. 5b). Surface topology of Zn electrodes was characterized using 3D laser scanning confocal microscopy (LCSM) after 200 h of cycling. The bare Zn anode exhibited peaks and valleys, with a significantly rougher surface (Ra: 344.9 μm) (Fig. 5c). These disorderly dendrites increased the surface area of the electrolyte/Zn foil, further accelerating the HER. However, for the 3D-LFGC@Zn and 3D-RFGC@Zn anodes, the roughness only showed slight changes (Ra: 56.2 μm and 36.7 μm) after repeated cycling (Fig. 5d–e). Additionally, SEM images revealed that on the bare Zn foil, scattered and loose Zn grains coexisted with large Zn flakes unevenly distributed, and some Zn particles penetrated into the glass fiber separator, indicating severe Zn dendrite growth (Fig. 5f). By contrast, a dense and dendrites-free surface was presented for 3D-RFGC matrix after cycles, as evidenced in Fig. 5g, h. Zn corrosion is analyzed by linear polarization measurements in 2 M ZnSO$_4$ electrolyte, where the corrosion potential of 3D-RFGC@Zn (−0.958 V), 3D-LFGC@Zn (−0.983 V) was higher than the bare Zn electrode (−1.014 V) (Fig. 5i). Moreover, after depositing/stripping certain cycles in ZnSO$_4$ electrolyte, massive by-products, Zn$_4$SO$_4$(OH)$_6$·xH$_2$O were generated due to severe corrosion for bare Zn electrode, whereas there are nearly no by-products on 3D-RFGC@Zn electrode according to the results of ex-situ XRD patterns (Fig. 5j)[51–53], which indicates that the 3D hierarchical graphene matrices can protect Zn anode from further corrosion. The in situ electrochemical impedance spectroscopy (EIS) technique was utilized to investigate the evolution of interfacial transport kinetics of the electrodes during continuous Zn plating and stripping. For bare Zn electrode, the charge transfer

resistance ($R_{ct}$) of initial electrochemical impedance is 622.3 Ω (Fig. S28, Table S3), which is larger than that of in the modified 3D-RFGC@Zn electrode (182.54 Ω) (Fig. 5k, Tabel S3). Besides, the $R_{ct}$ of bare Zn anode would gradually increase to be 782.6, 1008.6, 4194.4, and 8712.5 Ω at 6th, 10th, 30th, and 50th, respectively. However, the $R_{ct}$ of 3D-RFGC@Zn electrode initially decreases and tends to be stable in the subsequent record cycles, indicating the efficient interfacial transport kinetics. These achievements demonstrate that 3D-RFGC matrices can guide uniform and dense Zn plating together with high reversibility, thus suppressing Zn corrosion ability.

To gain further insights into the nature of the stable electrode-electrolyte interface, X-ray photoelectron spectroscopy (XPS) tests were performed at various sputtering depths to reveal the distribution of 3D elements in the solid electrolyte interface (SEI) layer (Fig. 6a–d). The binding energies were calibrated using the C 1s peak (at 284.8 eV) as the reference. The Zn metal peaks were clearly observed during continuous etching. The O 1s spectrum displayed two main components: SO$_4^{2-}$ at 532.1 eV and ZnO/Zn(OH)$_2$ at 530.4 eV, which can be attributed to the precipitation of Zn salt in the ZnSO$_4$ electrolyte[54,55]. The S 2p spectrum further confirmed the presence of ZnSO$_4$ (169.4 eV) along with minor ZnS species at 162.1 eV. As the sputtering time increased, the intensity of the C 1s, O 1s, and S 2p signals gradually decreased without any noticeable shifts, indicating the uniformity of the SEI film on the 3D-RFGC@Zn electrode. To verify the deposition/stripping of Zn in the 3D-RC, 3D-LC, 3D-RFGC, and 3D-LFGC matrices, we further simulated the Zn$^{2+}$ ion concentration and current density distribution using COMSOL Multiphysics (Fig. S29). As shown in Fig. 6e, the Zn$^{2+}$ ion concentrations of the internal and external space of the 3D-RFGC matrices are more uniform and lower than that of the 3D-RC as the reaction proceeds (Fig. S30), indicating homogeneous Zn plating in the inner space. When the reaction reaches 120 s, the Zn$^{2+}$ ions inside the 3D-RFGC matrices are largely consumed, which makes the ion concentration distribution more uniform from the top views (Movie S1). Without the introduction of GFs and VGs, it is obvious that the zinc-ion concentration on the upside of the 3D-RC tends to decrease as time increases, which indicates that the zinc ions are more inclined to deposit on the top surface than other sites (Fig. S31, Movie S2). Likewise, the 3D-LFGC demonstrates faster and more uniform Zn$^{2+}$ ion concentrations of the internal and external channels of the 3D-LFGC than that of 3D-LC (Figs. S32 and S33). The 3D-LFGC matrices with abundant N-doping sites can effectively regulate the deposition site of Zn and reaction areas. The spatial confinement effect usually exists in the nucleation of zinc metal in the channels, which is beneficial to suppress the formation of zinc dendrites. Bottom-up deposition of Zn occurs in 3D-LFGC matrices due to the layers of GFs and VGs (Movie S3, and Movie S4). However, the difference in electrode thickness results in the complete stripping of zinc metal. Therefore, the deposition and stripping efficiency of zinc metal will fluctuate greatly under the test conditions of high rates and areal capacities. Meanwhile, the local current density distribution was simulated to predict the zinc plating rate among carbon matrices. It is shown that the current density on the top surface of the 3D-RC matrices was larger than that of the 3D-RFGC matrices (Fig. 6g, S34), which indicates that the zinc plating rate on the top surface of the 3D-RC matrices was much higher than the 3D-RFGC matrices. The high local current density was forced to grow Zn dendrites on the upside of the bare carbon matrices (Movie S5). Similarly, the 3D-LFGC possesses a lower local current density than the 3D-LC, which contributes to the uniform deposition of Zn metal (Fig. S35, Movie S6). Compared with the 3D-LFGC matrices, the 3D-RFGC matrices show more open space to realize adequate Zn plating/stripping process due to the short Zn$^{2+}$ ions diffusion pathways and rich contact sites. SEM images reveal that uniform and dense Zn under high current densities and capacities can be deposited on 3D-RFGC matrices (Fig. 6i) and 3D-LFGC matrices (Fig. 6j), which are ascribed to uniform Zn$^{2+}$ ion concentrations and

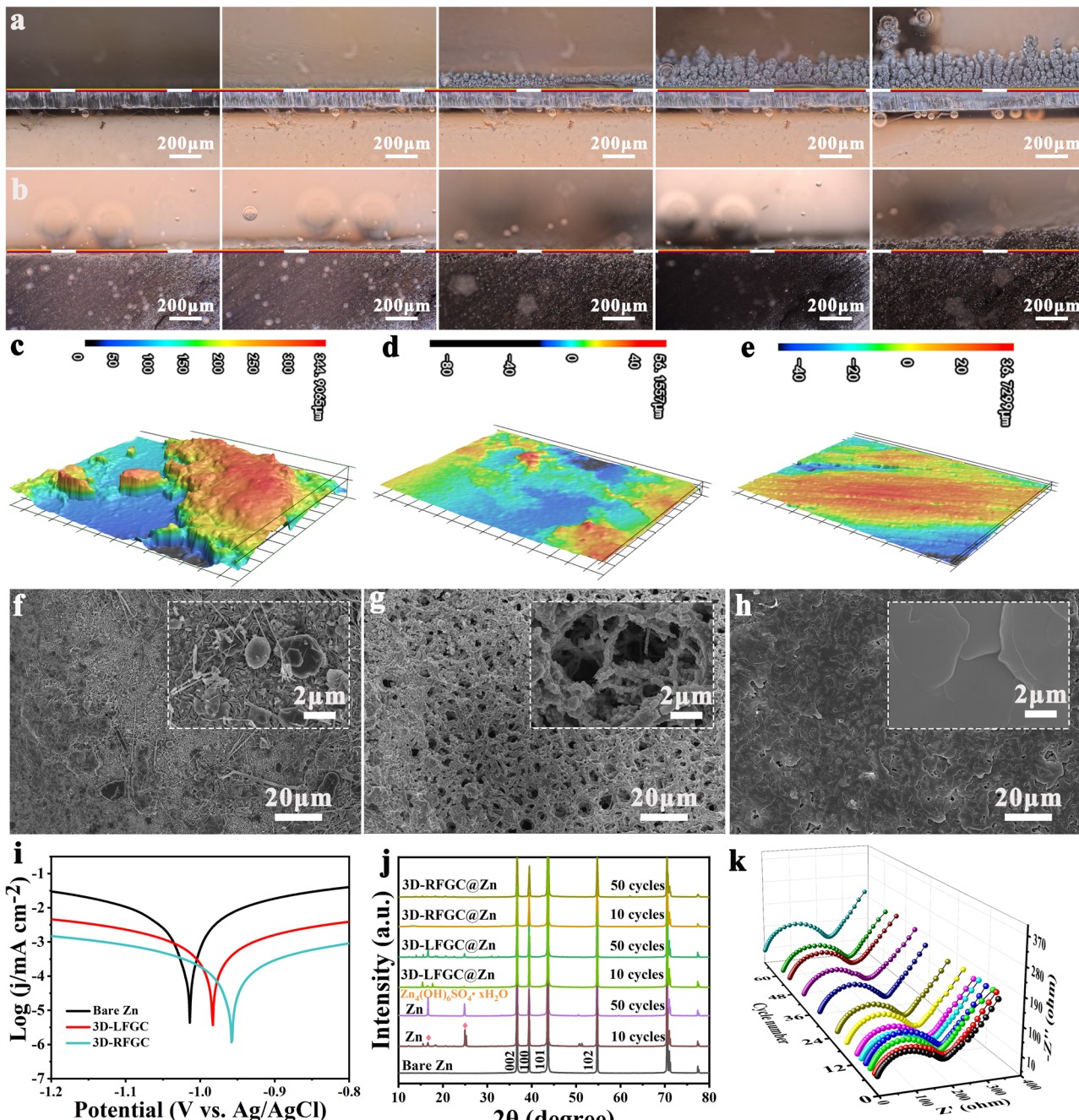

**Fig. 5 | Exploration of Zn deposition mechanism.** In situ observation of Zn deposited on the **a** bare Zn anode and **b** 3D-RFGC@Zn anode at the current density of 40 mA cm⁻². LCSM images of **c** bare Zn, **d** 3D-LFGC@Zn, and **e** 3D-RFGC@Zn anodes after 200 h cycling; **f**–**h** corresponding to the SEM images. **i** Linear polarization curves for describing corrosion of various Zn anodes. **j** Ex-situ XRD patterns of bare Zn, 3D-LFGC@Zn, and 3D-RFGC@Zn collected at different cycles. **k** In situ EIS curves of 3D-RFGC@Zn anode symmetric cells during continuous Zn plating/stripping process.

reduced local current density. The electrochemically active surface areas (ECSAs) of 3D-LC, 3D-RC, 3D-LFGC, and 3D-RFGC electrodes were estimated according to an electrochemical double-layer capacitance (DLC) method[56,57]. Figure 6k illustrates that the double-layer capacitance (Cdl) of the pristine 3D-LC electrode is considerably lower (1.04 mF cm²) compared to the pristine 3D-RC electrode (2.5 mF cm²). However, upon the growth of GFs and VGs, the Cdl of the resulting 3D-RFGC electrode is significantly increased to 15.7 mF cm² (Fig. S36), surpassing that of 3D-LFGC (12.51 mF cm²). Clearly, the 3D-LC and 3D-RC show an ECSA of 26 and 62.5 cm², respectively. The ECSA of the 3D-

RFGC matrix increases to 392.5 cm², higher than that of the 3D-RFGC matrix (312.75 cm²) (Fig. 6l). The comparable ECSA for 3D-RFGC can be ascribed to the introduction of GFs and VGs with enlarged specific surface area and abundant active sites. Larger ECSA may further facilitate Zn²⁺ diffusion and increase $D_{Zn}^{2+}$ (Zn²⁺ ions diffusion coefficient). Therefore, both theoretical simulation and experiment proved that the 3D-RFGC matrices can realize uniform plating of Zn metal and maintain a stable electrolyte-electrode interface during deposition and stripping, so as to achieve low impedance values, rapid ion transport and high space utilization of host surface.

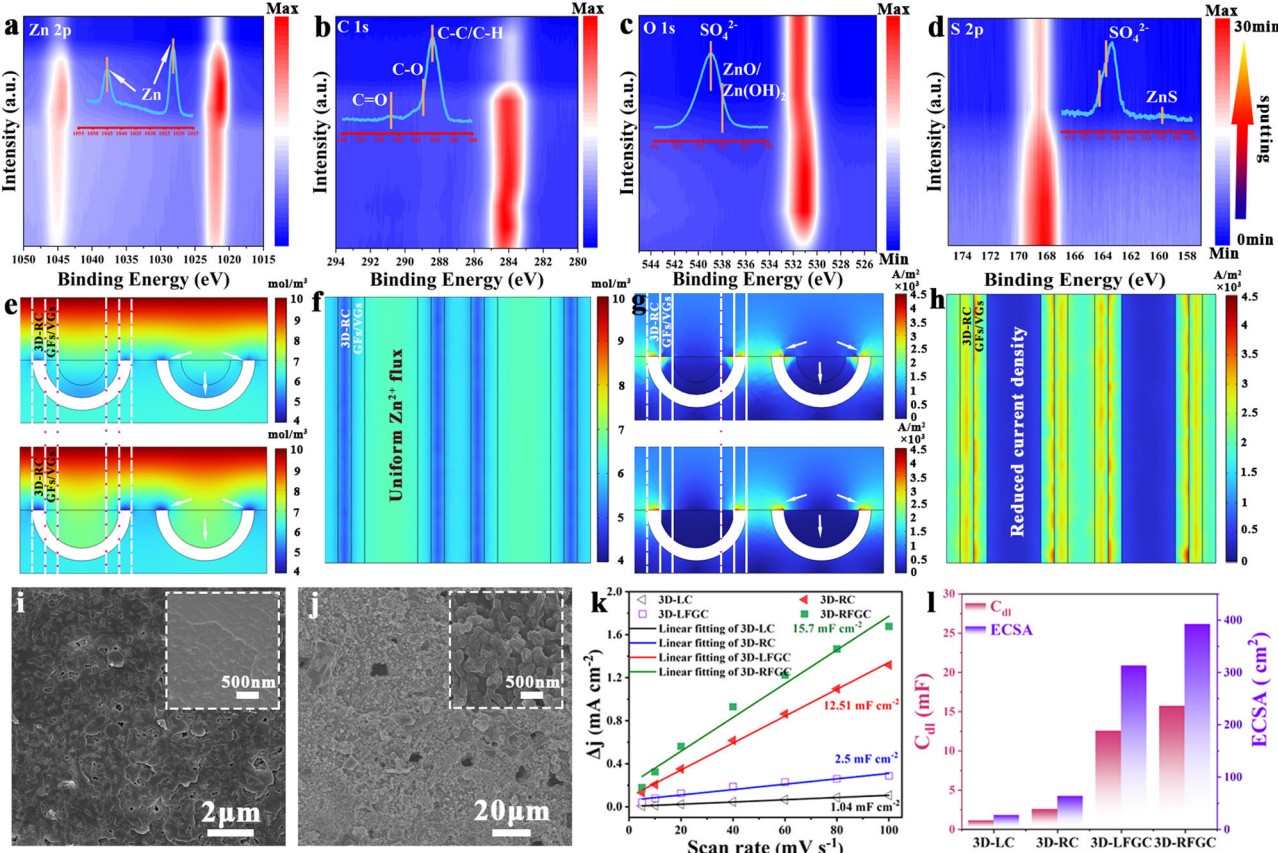

**Fig. 6 | Investigation of the interphase component and transport kinetics in the electrode-electrolyte interface. a** XPS depth profiles of **a** Zn *2p*, **b** C *1 s*, **c** O *1 s*, and **d** S *2p* for the 3D-RFGC@Zn anode with various Ar sputtering times. Simulation results of the Zn²⁺ ions concentration distribution for **e** 3D-RFGC@Zn and 3D-RC@Zn anodes; **f** the corresponding top-view information; **g** simulation results of the local current density distribution for 3D-RFGC@Zn and 3D-RC@Zn anodes;

**h** the corresponding top-view information of 3D-RFGC@Zn electrode. SEM images of **i** 3D-RFGC and **j** 3D-LFGC matrices after depositing Zn (The corresponding insets are SEM images of the magnified region); **k** the linear fitting of Δj (difference between anodic and cathodic current densities) versus scan rate for ECSA estimations of the various electrodes, and **l** quantitative comparison of double-layer capacitance and calculated ECSA results.

## Electrochemical properties of 3D-RFGC@Zn anode coupled with different 3D cathodes

In order to demonstrate the feasibility of the 3D hierarchical graphene matrices in a practical battery system, full cells were assembled using 3D matrices for both Zn anode and V₂O₅ (MnO₂, AC) as illustrated in Fig. 7a. By vacuum filtration, high loading V₂O₅@3D-LC cathodes between 1.0 mg cm⁻² and 36 mg cm⁻² were achieved (Fig. S37). SEM images and EDS mapping reveal that V₂O₅ particles are uniformly filled in the 3D channels (Fig. 7b, S38, and S39). The V₂O₅@3D-LC/3D-RFGC@Zn full cells achieved a capacity of 256.55, 251.53, 199.13, and 148.06 mAh g⁻¹ at rates of 2.0, 4.0, 20.0, and 40.0 mA cm⁻², respectively, as displayed in Fig. 7c. At 40.0 mA cm⁻², the V₂O₅@3D-LC/3D-RFGC@Zn delivers a high capacity of 156.2 mAh g⁻¹ (Fig. 7d). As a comparison, the discharge capacity of the V₂O₅//Zn cell quickly decreases to only 82.7 mAh g⁻¹ at 40.0 mA cm⁻² (Fig. S40). The improvement in the electrodynamics of Zn²⁺ ions transport can be further confirmed by in situ EIS (Fig. 7e, Table S4). A small charge transfer resistance (Rct) for the V₂O₅@3D-LC//3D-RFGC@Zn full cell (200.61 Ω) can be observed in the first cycle, and the Rct gradually decreases to 94.46, 22.18, and 16.91 Ω at the 8th, 20th, and 27th cycle, respectively. Such small Rct suggests lower charge transfer resistance and fast Zn-ion diffusion kinetics. The V₂O₅@3D-LC cathode with different loadings exhibits a hopeful areal capacity of 0.828, 4.87, 12.0, and 16.69 mAh cm⁻², as depicted in Fig. 7f. Such a remarkably high areal capacity of cathode is superior to most previous studies so far, which is ascribed to 3D graphene matrices with high conductivity and fast ion transport channels.

The MnO₂@3D-LC/3D-RFGC@Zn cells were also assembled and tested. Manganese dioxide nanowires (α-MnO₂) were synthesized as cathode materials for zinc-ion batteries by a hydrothermal method[58]. The XRD pattern of the as-prepared MnO₂ nanowires matches well with the standard phase of α-MnO₂ (JCPDS: 44-0141) (Fig. S41a). SEM images show a uniform one-dimensional nanowire structure with an average diameter of 100 nm (Fig. S41b). The MnO₂@3D-LC/3D-RFGC@Zn full cell exhibits significantly enhanced cycling stability, with a capacity retention of 99.1% for 500 cycles (Fig. 7g, S42), confirming the benefits of the 3D-LC matrices for stabilizing the Zn metal and MnO₂ cathode material thus achieving high-performance AZIBs. Meanwhile, aqueous Zn-ion hybrid capacitors (ZHCs) were assembled with 2.0 M ZnSO₄ solution as the electrolyte. The cyclic voltammetry curve (CV) of ZHCs with 3D-RFGC@Zn anode at different scan rates indicates excellent electrochemical reversibility and stability (Fig. 7h). The nearly rectangular outline shows that the AC@3D-LC/3D-RFGC@Zn has a typical capacitive behavior. The AC@3D-LC/3D-RFGC@Zn presents superior cycling and rate capability at different current densities (Fig. 7i, S44). In detail, the AC-3D-LC/3D-RFGC@Zn capacitor delivers the specific capacity of 256, 152.5, and 123.5 mAh g⁻¹ at the current densities of 2.0, 60.0, and 120.0 mA cm⁻². The cycling sustainability measurement reveals an excellent stability for 20000 cycles with a capacity retention of 75% at a current density of 40.0 mA cm⁻² (Fig. 7j). All in all, for Zn anode, the 3D-RFGC matrices consisted of GFs clusters and VGs possessing large specific surface area and efficient conductive network, which contributes to minimizing

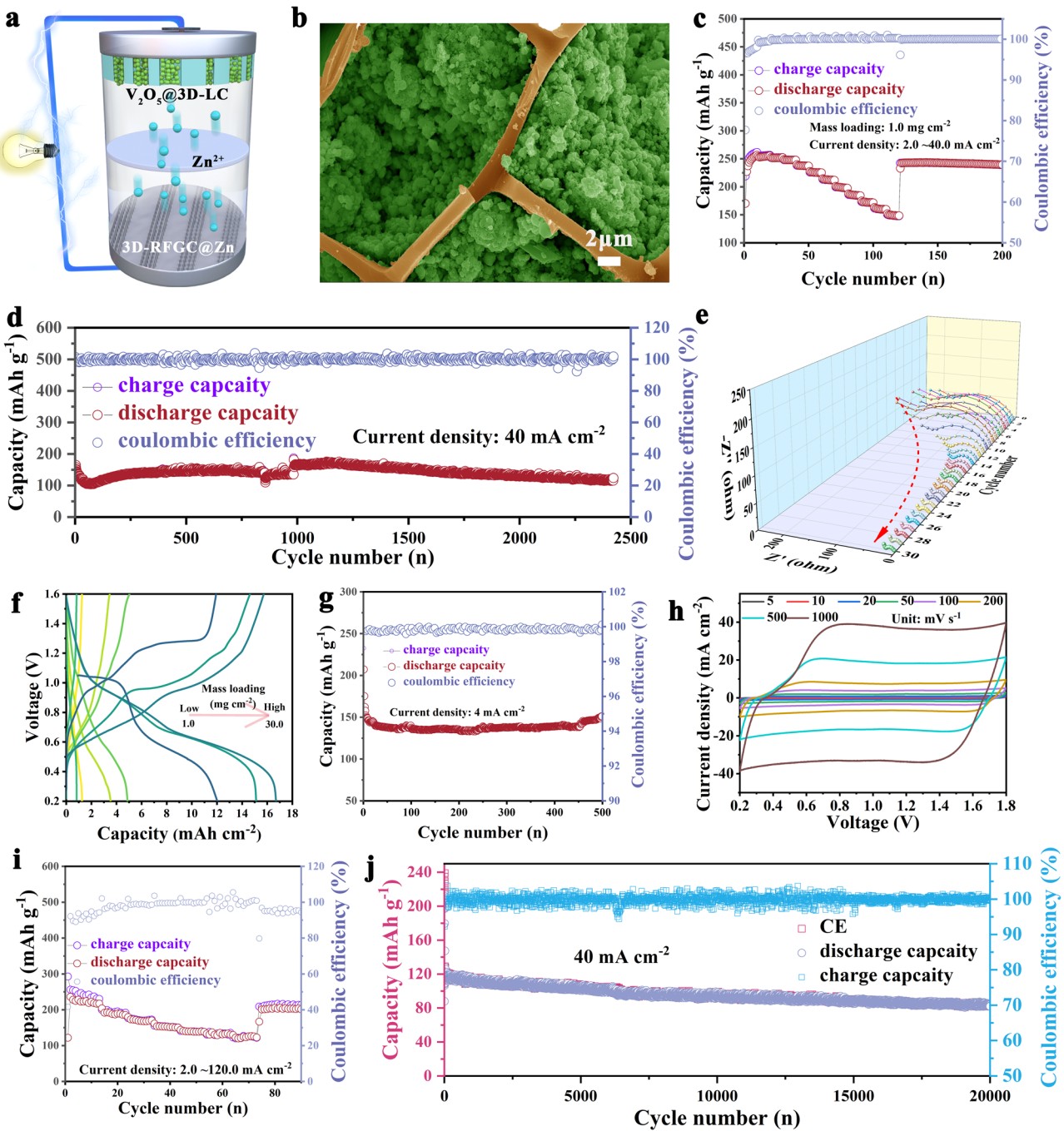

**Fig. 7 | Electrochemical performance of 3D-RFGC@Zn anode used in full cell and capacitor. a** The structural design of full cells using 3D matrices for both anode and cathode. **b** SEM image of 3D cathode. **c** Rate capability, **d** cycle performance at 40 mA cm⁻², **e** EIS curves, and **f** areal capacities under different loadings of V₂O₅@3D-LC/3D-RFGC@Zn. **g** Long-term lifespan of MnO₂@3D-LC/3D-RFGC@Zn full cells. **h** CV curves, **i** rate performance, and **j** cycling capability of the AC@3D-LC/3D-RFGC@Zn capacitor at 40 mA cm⁻².

local current density and accelerating the conduction of Zn²⁺ ions, thus achieving uniform Zn deposition. The 3D hierarchical graphene structure with enough spaces and abundant zincophilic sites is instrumental in storing high Zn capacity and relieving the volume changes during the plating/stripping processes (Fig. S44). For 3D V₂O₅ or α-MnO₂ and AC cathode, the entire 3D graphene skeleton makes cathode electrodes possess a high conductive network, and the rich porous channels facilitate the rapid transport of Zn²⁺ ions. The remarkable porosity and zincophility increase the wettability and storage of the electrolyte. Therefore, the 3D graphene matrices we

designed can inhibit the issues of dendrite growth of Zn anode and the trouble of cycling stability of cathode electrodes in AZIBs.

## Discussion
In summary, we present a novel approach utilizing free-standing, lightweight, and zincophilic 3D hierarchical graphene matrices, comprising N-doped GF clusters, VGs, and multichannel carbon matrices, to construct high-performance and stable Zn composite anodes with exceptional rates and capacities. The 3D-RFGC matrices, synthesized via a one-step thermal CVD process, exhibit high surface area, porosity,

and uniform porous structure, effectively mitigating local current density, reducing $Zn^{2+}$ ion concentration gradient, and ensuring a homogenized electric field distribution to regulate Zn deposition. Surface modifications of vertical graphene arrays (VGs) and graphene nanofiber clusters (GFs) with abundant zincophilic sites promote the realization of a high-performance Zn metal anode with both high current density and surface areal capacity. As a result, the engineered 3D-RFGC@Zn anode demonstrates an impressive CE of 99.67% over 3000 cycles with lower overpotential at a high current density of 120 mA cm$^{-2}$. The symmetric cell employing the 3D-RFGC@Zn anode exhibits excellent stability over 7200 cycles, featuring flat deposition and fast kinetics. Especially notable is the outstanding performance of the 3D-RFGC@Zn anode at a current density of 80 mA cm$^{-2}$, where it operates stably with an ultrahigh capacity of 80 mAh cm$^{-2}$ over 2400 h. Furthermore, when incorporated into full cells such as $V_2O_5$@3D-LC/3D-RFGC@Zn, $MnO_2$@3D-LC/3D-RFGC@Zn, and AC@3D-LC/3D-RFGC@Zn capacitor, these 3D graphene matrices contribute to excellent rate performance and remarkably improved cycling stability, owing to their ability to facilitate homogeneous Zn deposition. This strategy of employing 3D graphene matrices as Zn anodes presents a promising pathway for developing metal anodes capable of operating under high rates, capacities, and depth-of-discharge conditions.

## Methods

### Synthesis of 3D-LFGC and 3D-RFGC matrices

The 3D-LFGC and 3D-RFGC matrices were prepared via thermal-chemical vapor deposition (T-CVD) technology for growing vertical graphene arrays (VGs) and graphene nanofibers clusters (GFs) on the multichannel carbon matrices (3D-LC and 3D-RC). At 1050 °C, the methane (CH$_4$, 20–60 mL min$^{-1}$) used as a carbon source was introduced into the quartz furnace when the temperature reached 1050 °C. The VGs and GFs can be grown successfully between 1050 °C on the original carbon matrices by introducing H$_2$ (80-120 mL min$^{-1}$). The concentrations of H$_2$ and CH$_4$ varied with the temperature and played a crucial role in growing graphene nanofibers. The VGs heights can be controlled by varying the growth time.

The origin of the GFs was investigated. The growth of vertical graphene on the surface of pure carbon-based materials such as carbon black particles, carbon fiber (CF), and carbon nanofiber (CNF) was compared, and found that only vertical graphene sheets exist on the surface of the above-mentioned matrix, and there are no clusters of graphene fibers. Further, by verifying the catalytic effect by immersing the wood in an aqueous solution of ferric nitrate (Fe(NO$_3$)$_3$·9H$_2$O), we found that the nanofibers clusters grew more rapidly in the presence of the catalyst, this further verifies our hypothesis of catalytic effect. This provides ideas for the rapid preparation of gaseous carbon nanofiber composites.

### Synthesis of the composite 3D Zn anodes

The 3D-LFGC@Zn and 3D-LFGC@Zn composite anodes were prepared through the electrochemical deposition method. Zinc foil (150 μm) was used as the zinc source. By assembling the coin cell, Zn metal was deposited on 3D-LFGC and 3D-LFGC matrices. During the deposition process, the current density was selected to be 1 ~ 10 mA cm$^{-2}$, and the capacity was between 1 and 120 mAh cm$^{-2}$. To improve the preparation efficiency of the composite 3D Zn electrode, we usually choose a larger current to deposit to shorten the deposition time.

### Synthesis of $V_2O_5$@3D-LC, $MnO_2$@3D-LC, AC@3D-LC composite cathodes

The $V_2O_5$@3D-LC composite cathode was obtained through a vacuum filtration method. First, commercial cathode $V_2O_5$ material was dispersed in NMP solvent to make the slurry. Second, the 3D-LC host was placed on a vacuum apparatus, and added the slurry was drop by drop. The process lasted 1 min-30 min to make the cathode slurry infiltrate

into the channels of the 3D-LC. Third, the samples of as-infiltrated were kept in a vacuum oven at 80 °C for 12 h, and $V_2O_5$@3D-LC composite cathode was obtained. The mass loadings of $V_2O_5$ were calculated by direct weighting, which was adjustable by varying the infiltration time and controlling the total amount of slurry dropping. In this work, the mass loadings of the $V_2O_5$ cathode were controlled between 1.0 and 30 mg cm$^{-2}$. Similarly, $MnO_2$@3D-LC and AC@3D-LC cathodes were obtained by the same method.

Two points need to be emphasized. First, to load cathode materials, 3D-LC is considered the best 3D host for cathode materials in 3D-LC, 3D-RC, 3D-LFGC, and 3D-RFGC, because of its large space and connected macroporous structure. In detail, 3D-RC is difficult to filter due to its zigzag pore structure in the radial direction, and the active materials are attached to the upper area of the 3D host. The introduction of GFs also increases the difficulty of the filtration process. Second, the Zn half cells and full cells were assembled in CR2032 coin cells. In symmetrical cells tests, the bare Zn foil is a small circle with a diameter of 10 mm, and the sizes of 3D-RC, 3D-LC, 3D-RFGC, and 3D-LFGC are about 5 × 5 × 0.3 mm, 5 × 5 × 0.5 mm, 5 × 5 × 0.3 mm, and 5 × 5 × 0.5 mm (length × width × height), respectively. In full cells, the dimensions of the 3D host for cathodes in full cells are -5 × 5 × 0.5 (length×width×height), and the dimensions of the 3D host for anode in full cells are ~ 8 × 8 × 0.3 (length × width × height).

### Synthesis of $MnO_2$ nanowire cathode

To prepare the $MnO_2$ nanowire cathode via the hydrothermal method, initially, 3 mmol of MnSO$_4$·H$_2$O and 2 mL of 0.5 mol L$^{-1}$ H$_2$SO$_4$ were dissolved in 60 mL of deionized (DI) water and stirred for 20 min. Subsequently, 20 mL of 0.1 mol L$^{-1}$ KMnO$_4$ solution was added to the mixture. The resulting solution was vigorously stirred at room temperature for 1 h. Next, the mixture was transferred into a 100 mL Teflon-lined autoclave and sealed, followed by incubation at 120 °C for 12 h. After cooling, the obtained product was collected by centrifugation and washed three times with DI water. Finally, the product was dried using a freeze-dryer.

### Materials characterizations

The morphologies were characterized by Hitachi SU-8230 field emission scanning electron microscopy (SEM, Hitachi SU-8230). TEM, energy-dispersive X-ray analysis (EDX), and elemental mapping were performed using a Talos instrument with an acceleration voltage of 300 kV. X-ray diffraction (XRD, Bruker Advance D8, Ultima IV with D/teX Ultra with Cu-Kα radiation) was employed to characterize the crystalline structures of samples with a scanning rate of 5° min$^{-1}$. X-ray photoelectron spectra (XPS, Escalab 250Xi) were acquired on a Thermo SCIENTIFIC ESCALAB 250Xi with Al Kα (hυ = 1486.8 eV) as the excitation source. Raman spectra were performed on a HORIBA Lab-RAM HR Evolution using a 532 nm laser as the excitation source. The micro-CT was detected by Diondo D2 micro-CT scan system.

### In situ optical microscope characterization

The cells used for in situ optical microscope observations were assembled using molds obtained from Beijing Scistar Technology Co., Ltd. The working electrodes consisted of the 3D-FGC host and Zn foil, respectively. Zn metal was used as the counter electrode. The electrolyte used was a 2 M ZnSO$_4$ solution dissolved in deionized water (DI). An electrochemical workstation was employed to supply the power, and a current density of 40 mA cm$^{-2}$ was applied.

### Electrochemical performance assessment

To assess the electrochemical behavior and Coulombic efficiency (CE) of Zn plating and stripping, CR2032-type coin cells were assembled using Cu foil, Zn foil, 3D-LC, 3D-RC, 3D-LFGC, and 3D-RFGC as working electrodes. Zn metal served as the counter/reference electrode, and a glass fiber separator was used. The electrolyte employed was a 2 M

$ZnSO_4$ solution dissolved in deionized water. The electrochemical performance was evaluated using a Neware battery test system (Shenzhen, China) at a temperature of 30 °C. For CE evaluation, Zn was plated onto the various substrates to a capacity of 1 mAh cm$^{-2}$ (ranging from 1 to 120 mAh cm$^{-2}$) and subsequently charged to 0.6 V to strip the Zn in each cycle at different current densities (ranging from 1 to 80 mAh cm$^{-2}$).

To assess cycling stability and voltage hysteresis, symmetric cells were assembled using Zn/Zn, 3D Zn foam/3D Zn foam, 3D-LFGC@Zn, and 3D-RFGC@Zn composite anodes. These cells were cycled at various current densities (1-80 mA cm$^{-2}$) with different capacities of Zn deposition (1-80 mA cm$^{-2}$). Galvanostatic cycling was performed during Li plating/stripping while recording the potential over time. Electrochemical impedance spectra (EIS) measurements were conducted using a CHI 760D electrochemical workstation with a frequency range of 100 kHz to 100 mHz (CHI 760D, Shanghai CH Instruments Co., China).

### Full cell assessment

For $V_2O_5$-based full cells, the cathode electrodes were obtained by infiltrating commercial $V_2O_5$ cathode powders into the 3D-LC host. The free-standing cathode was obtained after removing the solvent from the slurry. For comparison, $V_2O_5$, conductive additives (Super P), and polyvinylidene fluoride (PVDF) were added to N-methylpyrrolidone (NMP) with a mass ratio of 7:2:1, and the mixture was stirred for 4 h to obtain a slurry which was then blade-coated onto titanium foil. For $V_2O_5$ cathode, the cycling curves were measured at 40 mA cm$^{-2}$, and rate curves were tested at 2–40 mA cm$^{-2}$ in the voltage window of 0.2–1.6 V. The 3D-RFGC@Zn anode was obtained by electrochemical deposition with a fixed capacity of 40 mAh cm$^{-2}$.

For $MnO_2$@3D-LC nanowire cathode, the electrodes were prepared by infiltrating method. The cycling curves were measured at 4.0 mA cm$^{-2}$ in the voltage window of 1.0–1.8 V. All cells were tested at 30 °C.

For AC@3D-LC cathode, the electrodes were prepared by infiltrating method. The cycling curves were measured at 40.0 mA cm$^{-2}$, and rate curves were tested at 2-120 mA cm$^{-2}$ in the voltage window of 0.2–1.8 V. All cells were tested at 30 °C.

### Modeling for distribution of electric and zinc-ion

A 3D transient model is developed to numerically resolve coupled muti-physical including ion transport and chemical reactions.

### Governing equations

The conservation for the ion in the electrolyte domain concludes times a derivative term, electrimigration and diffusive transport, and ion consumption by chemical reactions.

$$\frac{\partial c_i}{\partial t} + \nabla \cdot \left(-D_i \nabla c_i\right) + \nabla \cdot \left(-z_i u_i F c_i \nabla V\right) = R_i \tag{1}$$

where $R_i = \frac{-i_{loc}}{2F}$ is the electrochemical reaction source term, $i_{loc}$ is the local volume current density, $D_i = \frac{1}{\varepsilon}D_0$ is the effective diffusion coefficient ($D_0 = 1 \times 10^{-6}$ cm$^2$/s is the intrinsic diffusion coefficient), $\varepsilon = 0.5$ is the porosity for porous electrode ($\varepsilon = 1$ for the electrolyte domain), $u_i$ is the mobility (defined by the Nernst-Einstein equation), $c_i$ is the ion concentration, $z_i = 2$ is the charge number, $F$ is the Faraday constant, and $V$ is the electric potential.

The Bulter-Volmer equation was used to describe the Zn plating process and the local current density was as a function of potential and ion ($Zn^+$) concentration:

$$i = i_0 \left(\exp\left(\frac{1.5F\eta}{RT}\right) - \frac{c_i}{c_i^0}\exp\left(-\frac{0.5F\eta}{RT}\right)\right) \tag{2}$$

where $i_0$ is exchange current density, $\eta$ is the overpotential, $c_i^0$ is the initial $Zn^+$ concentration. Note that the active specific surface area of the electrode is $10^{-9}$ m$^{-1}$.

### Boundary conditions

Figure S32 shows the boundary condition of two models. The initial concentration of $Zn^{2+}$ is 10 mol/m$^3$. The current density $i$ and $Zn^{2+}$ concentration at the top of the bulk solution is 100 mA/cm$^2$ and 10 mol/m$^3$, respectively. Note that all electrode surfaces and porous electrodes domains are the electrochemical reaction sites. The potential of the electrode surface is 0 V, and the rest boundaries are insolution. Besides, the rest boundaries for the ion transport are no flux.

### Reporting summary

Further information on research design is available in the Nature Portfolio Reporting Summary linked to this article.

## Data availability

All relevant data that support the findings of this study are presented in the article and Supplementary Information. Source data are provided in this paper.

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

## Acknowledgements

This work was financially supported by the Stable Support Plan Program for Higher Education Institutions (no. 20220815094504001) and Shenzhen Key Laboratory of Advanced Energy Storage (ZDSYS20220401141000001). The authors would also like to acknowledge the technical support from SUSTech Core Research Facilities.

## Author contributions

Y.M. and L.Z. designed this study. Y.M. synthesized the key materials, carried out the characterization, and Y.M., Z.L., and B.-K. W. performed the electrochemical measurements. H.H. performed COMSOL simulations. F.W., Y.C., and L.F.Z. analyzed partial data. M.Y. provided cathode materials. J.H. performed XRD measurement. L.Y. provided activated carbon materials used in Zn capacitor testing. M.H. revised the introduction section of the manuscript. Y.M. and L.Z. organized and wrote the manuscript with contributions from other authors. Y.M., Z.L., and B.-K.W. contributed equally to this work.

## Competing interests

The authors declare no competing interests.
