## [Peer Review File · Nature Communications]

3D hierarchical graphene matrices enables stable Zn anodes for aqueous Zn batteriesREVIEWER COMMENTS

Reviewer #1 (Remarks to the Author):

This manuscript describes the synthesis of sophisticated 3D carbon matrices as current collectors for zinc metal anode. The long-time cycling at 80 mA cm⁻²/80 mAh cm⁻² is impressive, at least on the surface. There are multiple flaws that, unless satisfactorily justified, will prevent the manuscript from publication in Nat. Commun.

- 1) There appears to be two carbon matrices, namely 3D-LFGC and 3D-RFGC, that the manuscript is reporting and comparing. However, there is only characterization and synthesis procedure for one general "3D-FGC".
- 2) While the 80 mAh cm⁻² is impressive as an areal capacity, it is also only a 17% utilization of the 800-micron-thick carbon matrices and thus a poor volumetric capacity. Comparison of such an impractical electrode with other planar electrodes would not be fair.
- 3) The authors appeared to implicate improved zincophilicity of the carbon matrices but failed to show experimental evidence.
- 4) The authors stated "enlarged specific surface area" as a weakness for 3D matrices, but went on to synthesize carbon matrices with high surface area anyway, and these matrices somehow do not come with any of the perceived disadvantages of 3D matrices such as aggravated corrosion of zinc and HER.
- 5) The cartoons in Figure 1 and 2 are missing legend.
- 6) Figure 5a-e are missing scale bars.
- 7) The authors claimed a preference for zinc ions to deposit on the top of a control sample with Figure S19 and Video S2, but neither display item shows vertical information. The items that do show vertical information for the same sample, Figure S21 and S22, do not show any difference between the top and the bottom.

Reviewer #2 (Remarks to the Author):

Here, Mu et al. report an 3D hierarchical graphene matrix layer deposited on the Zn anode as an efficient tool for the longitudinal deposition and an inhibitor for the zinc dendrite growth, leading to extended cycle life for the half and full cells. Overall, the prolonged cycle data and calculated simulation data are appealing but there seem to be some major fundamental inconsistencies that do not explain the kind of performance claimed and communicated here. Experimental details are also not obvious, making it difficult to interpret the results.

1. In the main text Page 8, line 2-5, the graphene has zincophilicity and enough space to store Zn. Then, what is the specific evidence or reference? There is BET data that the 3D-FGC is estimated to be 142 m²/g but it seems difficult to support the fact that it reduces the concentration gradient of Zn²⁺ ions and enables uniform impregnation of zinc ions. (pg 8, line 15-20)
2. In the Figure 4a, the symmetric cell data of 3D-RFGC at the current density 1 mA/cm² does not appear to be effective compared to other papers (Ref. Nat. Commun, 13, 7922 (2022) & Nat. Commun. 11, 3961 (2020)). The author should have to load another data that shows constant overpotential for the entire cycling. If not, the sudden increase in overpotential, which occurs even in the best data (about 2000 h ~2500 h in the 3D-RFGC and about 2300 h ~ 4500 h in the 3D-LFGC), would require further proof which can prove that it is not a short circuit.
3. There is excellent paper (Ref. Electrochimica Acta, 2022, 425, 140648) about coating Zr-MOF

on Zn metal anode. The authors should make note of them.

4. The electrochemical performance of different electrodes of bare Zn, 3D-LFGC matrix, and 3D-RFGC matrix are shown in the figure 3a-i. I don't know if bare Zn is an appropriate comparison. For example, there is a reference study that has achieved good results by transforming zinc metal into a 3D matrix structure (Ref. Adv. Energy Mater.2022, 12, 2102797). Rather than looking at the morphology of plating in bare Zn, which is a 2D structure, it is considered that a more effective comparison method is to look at the plating process by processing zinc metal like the 3D structure presented in this study. Also, in figure 3j, there is not any information about the cell you used in this overpotential data. It would be nice to additionally fill in the information of the cell used in this graph.

5. In the figure 2p, it seems that contact angle measurement has been conducted, but the electrolyte used during the experiment is not mentioned in the paper or SI. It would be nice to add information about electrolytes Looking at figure 2p, it seems that the contact angle analysis was conducted, but the electrolyte used during the experiment is not mentioned in the text or SI. It would be nice to add information about electrolytes to the graph or to the text. And it is stated that the wettability is increased due to uniform N doping, and it would be much better to understand if you explain in detail the evidence reference or mechanism that supports this.

6. In the main text Page 14, line 20-24, and figure 5j, you identified the new peak as the $Zn_4SO_4(OH)_6 \cdot 3H_2O$. Do you have any references to support this?

7. In the Supporting Information Page 2, materials section, it was written that $ZnSO_4$ (2M) and $ZnSO_4/MnSO_4$ (2M+0.5M) were used when assembling the coin cell. However, in the Supporting Information Page 5, in situ optical microscope characterization and electrochemical performance assessment sections, it was written that 2M $MnSO_4$ was used as the electrolyte. Please check which information is correct.

Reviewer #3 (Remarks to the Author):

The authors developed a 3D hierarchical graphene matrix consisting of wood-derived multichannel carbon, N-doped graphene nanofibers clusters, and vertical graphene arrays. The matrix worked well when serving as a substrate for Zn metal anodes. The electrochemical performances are excellent. However, some flaws and concerns also need to be addressed.

1. I suggest the authors add more schematics and text descriptions of 3D-LFGC and 3D-RFGC. For example, how were 3D-LFGC and 3D-RFGC cut from the original 3D-FGC? What are the characteristic morphologies of the 3D-LFGC and 3D-RFGC electrodes?

2. The details of the coin cell assembly are not clear. What are the final dimensions of the 3D-LFGC and 3D-RFGC electrodes? What is the electrolyte volume? How did the authors ensure the electrolyte wetting of the entire thick 3D electrode?

3. What are the electrochemically active surface areas of 3D-LFGC and 3D-RFGC?

4. How did the authors determine the voltage hysteresis? At which capacity did the authors record the voltage? Is the voltage hysteresis in this work the sum or the average of the anodic and cathodic processes? Please add these details to the supporting information. I also recommend adding voltage hysteresis in Table S1 and S2.

5. In Figures 3b and 3c, why are there sharp turning points in the curves? If the sampling intervals are too wide, the calculated Coulombic efficiencies might not be accurate.

6. In Figure 3j, the authors did not compare the data with 3D-LFGC.

7. In Figure 3l, are the labels for 3D-LFGC and 3D-RFGC reversed?

8. The reviewer suggests that the authors double-check the data of this work in Figure 3m.

9. In the supporting information of 3D-FGC@Zn synthesis, the authors state that "High areal capacities, such as 5 or 10 mAh cm⁻², were also conducted to increase the deposition rate". Did the authors mean high areal current densities such as 5 or 10 mA cm⁻²?

10. The data of bare Zn was not presented in Figure 4f.

11. The charge/discharge curves of 3D-RFGC@Zn symmetrical cells fluctuate significantly at all rates, especially in Figure S14. Also, the overpotential at 1 mA cm⁻² is much higher than those at

other rates, the overpotential at 10 mA cm⁻² is much lower than that at 5 mA cm⁻², and the overpotential at 60 mA cm⁻² is much lower than that at 30 mA cm⁻², which do not make sense. Please explain these findings.

12. How exactly does 3D hierarchical graphene matrix protect Zn anode from corrosion, given that the 3D hierarchical graphene matrix would increase the accessible area of Zn? By reducing dendrites?

13. The reviewer suggests a table summarizing the fitted EIS data.

14. It seems like the active surface of 3D-RFGC is just the top quasi-2D layer, which means the electrolyte and ions cannot penetrate the top layers to access the bottom.

15. The reviewer doubts the correctness of the model for 3D-LFGC. From the SEM images (Figure S5), the channel openings are much larger than the top surface.

16. Following question #15, if the worse performance of 3D-LFGC came from the more accessible top surface and less accessible bottom of channels, would it be better to choose a woodblock precursor with larger channel sizes? What kind of natural wood block did the authors choose and why?

17. Following question #16, the authors claim that a high channel of 800 μm is not conducive to the complete stripping of zinc metal. However, there are many successful works reported for thick 3D Zn anodes (Adv. Energy Mater. 2021, 11, 2003927; Science 2017, 356(6336), 415; etc.).

18. What are the differences between the models of carbonized wood with and without vertical graphene arrays and graphene nanofibers? The models given in this work are indistinguishable from each other.

19. What are the Zn anode's loadings for the full device tests? The authors used gravimetric current densities and capacities. Were they calculated based on the mass of active cathode materials? Please also specify the mass loading of active cathode materials. What are the corresponding areal current densities, areal capacities, and depths of discharge of the Zn anode?

20. For 3D V2O5 cathodes, it seems like V2O5 was loaded on 3D-LFGC. Please specify this in the main text and consider changing the name to V2O5@3D-LFGC. V2O5@3DG in the main text is ambiguous. Why didn't the authors load V2O5 on 3D-RFGC?

21. Were MnO₂ and AC cathodes also 3D? The authors didn't mention this in either the main text or supporting information.

Dear reviewers:

Thank you very much for your time and work on our manuscript. Your positive and constructive comments are important for the improvement of our manuscript. According to your comments, we improved the manuscript, which was marked with red color in the revised manuscript. The modifications in our manuscript are described below.

Responses to the comments by Reviewer #1

Reviewer #1: This manuscript describes the synthesis of sophisticated 3D carbon matrices as current collectors for zinc metal anode. The long-time cycling at $80 \text{ mA cm}^{-2}/80 \text{ mAh cm}^{-2}$ is impressive, at least on the surface. There are multiple flaws that, unless satisfactorily justified, will prevent the manuscript from publication in Nat. Commun.

Comment #1: *There appears to be two carbon matrices, namely 3D-LFGC and 3D-RFGC, that the manuscript is reporting and comparing. However, there is only characterization and synthesis procedure for one general "3D-FGC".*

Response: Thank you for your comments. We have added additional characterizations and experiments of the major carbon matrices in the revised manuscript. Here, we detailed described the synthesis and structural characterizations of the 3D-LC, 3D-RC, 3D-LFGC and 3D-RFGC, as follows:

(1) Synthesis of Graphene Arrays

In this manuscript, we for the first time developed a three-dimensional (3D) hierarchical graphene matrices consisting of nitrogen-doped graphene nanofibers clusters (GFs) anchored upon vertical graphene arrays (VGs) modified multichannel carbon (3D-LFGC/3D-RFGC). The schematic illustration of fabricating the 3D-LFGC/3D-RFGC matrices is shown in **Figure 1**.

Figure 1. The schematic illustration of fabricating the 3D-LFGC/3D-RFGC matrices.

As shown in **Figure 1**, the overall synthesis process of 3D-LFGC/3D-RFGC matrices consists of three steps: i) carbonization to obtain porous carbon channels; ii) chemical vapor deposition (CVD) growth of GFs and VGs and iii) post-treatment of ammonia (NH_3) atmosphere.

i) Synthesis of Multichannel Carbon Matrices

First, the natural basswood (Jiangxi Three-Wood Technology Co., Ltd.) was cut along the radial direction and longitudinal direction (**Figure 2**). Wood can be cut into different sizes, shapes and thicknesses (**Figures 3a-3c, and 4a-4c**). This work used square or round wood slices with a thickness of 0.6~0.8 mm (**Figure 3d**). Second, the wood slices were stabilized in the air at 260 °C for 6 h. Third, the carbonization process was conducted at 1100 °C for 6 h in ammonia (NH_3 , 50 mL min^{-1}). The carbonized wood was activated in a carbon dioxide (CO_2 , 100 mL min^{-1}) flow at 750 °C for 12 h to obtain the activated multichannel carbon matrices (**Figures 3e and 4d-4e**). The carbon matrices shown in **Figure 3** and **Figure 4** were named 3D-LC and 3D-RC, respectively.

Figure 2. Schematic diagram of the natural basswood processing.

Figure 3. The optical photographs of raw wood and after different treatments including a) wood from the tree; b) different sizes and shapes; c) different thicknesses; d) used in this work and e-g) wood-derived multichannel carbon (3D-LC) with different sizes.

Similarly, the 3D-RC was obtained by the same process except for the first step of the cutting process, shown in **Figure 4**. Also, different sizes, shapes, and thicknesses of wood slices were manufactured in this manuscript. During the carbonization process, the thickness and size of the wood chips were shrunk.

Figure 4. The optical photographs of raw wood and after different treatments including a) wood from the tree; b) cross-sectional image; c) different sizes and shapes; d) pre-oxidation and e) carbonized multichannel carbon (3D-RC).

ii) CVD growth of GFs and VGs and N-doping treatment

In this stage, we successfully grew clusters of graphene fibers (GFs) and vertical graphene arrays (VGs) on the two carbon substrates obtained above by thermal

chemical vapor deposition (thermal-CVD) technology. A complete growth process of GFs and VGs was established for the first time by controlling the growth temperature, atmosphere, and time. In particular, in this manuscript, we report the first finding that thermal-CVD technology can achieve synchronous growth of GFs and VGs by prolonging growth time and increasing the proportion of carbon sources (CH_4). The experimental process is shown below (**Figure 5**):

Figure 5. The device of growing GFs and VGs by a thermal-CVD method.

At 1050 °C, the methane (CH_4 , 20-60 mL min^{-1}) used as a carbon source was introduced into the quartz furnace when the temperature reached 1050 °C. The GFs and VGs can be grown successfully at 1050 °C on the pristine carbon matrices by introducing H_2 (80-120 mL min^{-1}). The concentrations of H_2 and CH_4 varied with the temperature. The VGs and GFs heights can be controlled by varying the growth time. The as-obtained graphene productions from 3D-LC and 3D-RC can be named 3D-LFGC and 3D-RFGC, respectively. After the growth, the methane and hydrogen gas were turned off, and the introduction of ammonia is to achieve nitrogen doping.

SEM images, TEM images, XRD patterns, Raman spectra, TGA curves, specific surface area and micro-CT were conducted to analyze the structure and composition of 3D-LC, 3D-RC, 3D-LFGC and 3D-RFGC. In the revised manuscript, we make a detailed supplement and analysis of the two carbon materials, as seen in the text of the manuscript and supporting information file. The above two structures are briefly analyzed and explained here.

Figure 6 indicates the basic characteristic of the multichannel carbon matrices for 3D-LC and 3D-RC from the SEM images; the micro-CT images also show the porous and continuous structure, which is beneficial to deposit Zn metal and loading cathode materials of ZIBs.

Figure 6. SEM and micro-CT images of 3D-LC and 3D-RC.

The morphologies of 3D-LFGC and 3D-RFGC are shown in **Figure 7** and **Figure 8**. At this stage, we regulated the concentration of the reaction atmosphere (Ar/CH₄/H₂) to find the best growth conditions and extended the growth time, thus obtaining GFs and VGs with different heights and densities. The VGs are firmly anchored on the inner walls of the carbon channels, while the GFs are grown vertically from the surface of the carbon matrices directly and are shrouded in the upper layer of the VGs arrays. We have investigated the origin of the GFs. Based on our previous studies (*Adv. Sci.* 2022, 9, 2203; *Carbon* 2021,173, 477e484; *Adv. Sci.* 2022, 9, 2104685), we compared the growth of vertical graphene on the surface of pure carbon-based materials such as carbon black particles, carbon fiber (CF) and carbon nanofiber (CNF), and found that only vertical graphene sheets exist the surface of the above-mentioned matrices, and there are no GFs (**Figure 9**). In this regard, EDX mapping analysis of the original carbonized wood revealed the presence of a variety of micronutrients in the original wood (**Figure 10**). Further, by verifying the catalytic effect by immersing the wood in an aqueous solution of ferric nitrate (Fe(NO₃)₃·9H₂O), we found that the nanofibers clusters grew more rapidly in the presence of the catalyst (**Figure 11**), this further verifies our hypothesis of catalytic effect (*Adv. Energy Mater.* 2011, 1, 1205-121; *Energy Environ. Sci.*, 2010, 3, 1286-1293). This provides ideas for the rapid preparation of gaseous carbon nanofiber composites.

Figure 7. SEM images of 3D-LFGC host for longitudinal direction when extended the growth time a) 2 h; b) 6 h; c) 9 h and d) 12 h.

Figure 8. SEM images of 3D-RFGC host for radial direction when extended the growth time a) 2 h; b) 6 h; c) 9 h and d) 12 h.

Figure 9. SEM images of vertical graphene grown on a) carbon black, b) carbon nanofibers and c) carbon fibers.

Figure 10. EDX mapping of carbonized wood including a) SEM image, b) C, c) N, d) O, e) Mg, f) Ca, g) Fe, h) Zn, i) Na elements. (The scale bar is 40μm)

Figure 11. SEM image of 3D-RFGC in the presence of a catalyst.

Meanwhile, for the electrochemical test, we added the corresponding tests for both 3D-LFGC and 3D-RFGC. All these data are presented in the revised manuscript and supporting information. For example, *Figures 2f, 2g, 2m, 2n, 2q, 2r, S3, S5, S8, S10, S12, S19, S23, S25*, etc.

Comment #2: *While the 80 mAh cm⁻² is impressive as an areal capacity, it is also only a 17% utilization of the 800-micron-thick carbon matrices and thus a poor volumetric capacity. Comparison of such an impractical electrode with other planar electrodes would not be fair.*

Response: Thank you for your concern about our poor volumetric capacity and

performance comparison. Here we make some explanations and changes, as follows:

(1) The thickness of electrodes

We are very sorry that we made an error in the description of electrode thickness. The thickness of pristine wood used in the thermal treatment is 400 μm ~800 μm . After pre-oxidizing and carbonizing, the thicknesses and sizes of electrodes decreased due to dehydration and shrinkage. The thickness of 3D-LFGC and 3D-RFGC used in electrochemical experiments is ~500 μm and ~300 μm in this manuscript, respectively. Another idea can be provided to dispel your worries. The annealed carbon matrices can be polished with 2500 grit sandpapers, and cleaned by ultrasonic washing with deionized water and ethanol, thus obtaining low-tortuosity carbon matrices. The 3D-LFGC and 3D-RFGC matrices with different thicknesses can be obtained shown in **Figure 12** and **Figure 13**.

Figure 12. The 3D-LC matrices with different thicknesses of a) 800 μm , b) 500 μm , c) 360 μm , d) 280 μm , e) 160 μm and f) 120 μm .

Figure 13. The 3D-RC matrices with different thicknesses of a) 800 μm , b) 550 μm , c) 450 μm , and d) 300 μm .

To enhance the energy density of rechargeable batteries, thick electrodes become a research highlight in recent years. As shown in **Figure 14**, most works of thick electrodes possess higher thicknesses beyond 500 μm . These results demonstrate that the thick electrode is promising for practical application in energy storage devices and encourage more investigations for thick electrodes.

Figure 13. The comparison of thick electrodes previously reported (*Adv. Energy Mater.* 2017, 7, 1700595; *Energy Storage Mater.* 2022, 51, 476-485; *Sci. Bull.* 2022, 67, 1253-1263; *Chem. Eng. J.* 2021, 414, 128767; *Nano Energy* 2018, 45, 203-209; *Nano Lett.* 2021, 21, 5896-5904; *Proc. Natl. Acad. Sci.* 2017, 114, 3584-3589; *Nat Commun.* 2021, 12, 4519).

(2) Volumetric capacity of thick-electrodes

Mass specific capacity and volume specific capacity, as the two key indexes of the battery, are very important to explain the performance and application potential. However, it is also a challenge in practice to meet the two indexes simultaneously, especially for thick electrodes. However, traditional thick electrodes often lead to sluggish ion transfer kinetics as well as decreased electronic conductivity and mechanical stability, leading to their thickness-dependent electrochemical performance. In this manuscript, unfortunately, we do have the shortcomings, as you mentioned, poor volumetric capacity, but our work is of great significance in developing multi-function graphene matrices, realizing high current density/capacity Zn anode, and accelerating ion transport at thick electrodes. In other words, the innovation of this manuscript includes three points:

i) N-doped graphene nanofibers clusters (GFs) and vertical graphene arrays (VGs) modified multichannel carbon (3D-FGC) were for the first time achieved by a one-step thermal CVD method, providing ultrafast, continuous, and smooth electron transport channels. This unique 3D architecture with zincophilic sites exhibits a uniform Zn deposition, together with improved electrochemical performance.

ii) Ultrahigh current densities and capacities were demonstrated for the 3D-RFGC@Zn composite anode, exhibiting outstanding cycling stability and kinetics at a high rate and areal capacity, which provides a new road to achieve high-performance Zn batteries.

iii) Inspired by the thick electrodes, the freestanding $V_2O_5@3D-LC$, $MnO_2@3D-LC$, $AC@3D-LC$ cathodes with conspicuous mass loading matched with 3D-RFGC@Zn anode indicated excellent areal capacities.

(3) Performance comparison

According to your comments, we have modified the contents of *Figure 3l* and *Figures 4k-4l* in the manuscript, particularly with respect to the 3D hosts used in zinc anodes, which are presented in the manuscript as follows (**Figure 14 and Table 1**).

Figure 14. a) The comparison of CE, current densities of deposition, capacities and cycle capability in half-cells; b) the comparison of voltage hysteresis and current densities, areal capacities and cycle capability in symmetrical cells with previously reported 3D Zn anodes literature (The area of the circle represents the size of the areal capacity).

Table 1 Comparison of electrochemical performances of our 3D-RFGC@Zn anode and reported materials for ZMAs. (C_d : current density; C: capacity; L:lifespan, V_h : voltage hysteresis)

Samples	C_d (mA cm ⁻²)	C (mAh cm ⁻²)	L (h)	V_h (mV)	Reference
CuZIF-L@TM	6	7	650	80	14
TZNC	1	1	450	12	15
3D Ni-Zn	5	2	300	/	27
MXene/Graphene Aerogel	10	1	1050	64	16
CnC HS	4	1	116	40	17
3D Zn	10	1	190	45	18
3D Ti-TiO ₂	10	0.5	500	/	19
3DGT@Zn	2	1	1100	/	20

3D SnPCF	10	5	500	47	21
Zn/CNT	5	2.5	110	68	23
Cu-Ps/EG	10	10	3000	71.4	24
AgNWA@Zn//C-MOF	40	10	90	/	25
3D-Topo-Zn	2	2	1160	/	26
3D-ZGC	20	1	150	65	28
3DP-ZA	4	2	180	/	29
Zn@PCH	10	1	110		22
SS-V ₂ O ₃ @C/Zn	20	20	5000	/	30
3D-Zn	5	5	160	/	31
Ti ₃ C ₂ T _x	10	1	180	/	32
MXene/ZnS	1	1	70		33
N-VG@CC	10	5	650	43.5	34
3D-RFGC@Zn	1	1	7300	15	This work
3D-RFGC@Zn	20	20	2000	32	This work
3D-RFGC@Zn	40	40	2200	26	This work
3D-RFGC@Zn	80	80	2600	31	This work

Comment #3: *The authors appeared to implicate improved zincophilicity of the carbon matrices but failed to show experimental evidence.*

Response: According to your comment, we have supplemented contact angle measurements (CA) and density functional theory (DFT) theoretical simulations of the different carbon materials prepared in this manuscript, which are used in combination to account for enhanced zincophilicity. The experimental details and the discussion are summarized as follows:

(1) Contact angle measurements

Attributed to the abundant N-doped sites on GFs and VGs skeletons, the contact angle refers to the 2 M ZnSO₄ electrolyte significantly decreases to 7.92° for 3D-RFGC (**Figure 15**), implying it an enhanced electrolyte permeability. This ensures that even at high Zn deposition and high cathode electrode loading, the electrolyte can remain fully infiltrated, thus maintaining an efficient ion transport process. Specifically, we performed contact angle tests on bare Zn, 3D-LC, 3D-RC, 3D-LFGC, 3D-RFGC, 3D-

LFGC, and 3D-RFGC, respectively, and recorded the 20 s process by video. As shown in **Table 3** below, we concluded that the N-doped graphene matrices exhibit excellent wettability, which is consistent with the results reported in previous literature (*Adv Mater* 2020, 32, e2003425; *Nano Research* 2021, 15, 9785).

Figure 15. Optical images of contact angle on various current collectors in this work.

Table 3. The contact angle of bare Zn and various matrices used in this work.

Samples	1s	5s	10s	20s
Bare Zn	51.96	43.63	40.35	40.35
3D-LC	134.10	133.50	133.50	133.50
3D-RC	121.76	120.30	120.10	120.10
3D-LFGC	54.31	39.57	28.28	22.92
3D-RFGC	29.51	14.61	9.68	7.92

(2) Density functional theory (DFT) theoretical simulations

The introduction of zincophilic sites is critical in adjusting the absorbing/bonding ability of 3D hosts (*Adv Mater* 2020, 32, e2003425; *Adv Mater* 2021, 33, e2101649). We, therefore, conducted DFT computations to compare zincophilic ability of nitrogen and carbon in graphene models (**Figure 16**). The configurations of zinc adsorption on different sites on carbon substrates are illustrated in **Figure 17**. As shown in **Figure 18**, the binding energies of zinc-atom on different adsorption sites on graphene are, respectively, -0.0196 (Top), -0.0203 (Hollow), and -0.0180 (Bridge) eV. For the adsorption sites on N-doped graphene, the corresponding binding energies are,

respectively, -0.0246 (Top), -0.0270 (Hollow), and -0.0254 (Bridge) eV. This finding underscores that the introduction of nitrogen-doping changes the interaction of zinc and carbon substrate from zincophobic to zincophilic. Therefore, the 3D-LFGC/3D-RFGC matrices with zincophilic sites exhibited a uniform Zn deposition, together with boosted electrochemical performances.

Figure 16. Configurations of graphene (L.) and nitrogen-doped graphene (R.).

Figure 17. Configurations of zinc adsorption on different sites of graphene/nitrogen-doped graphene.

Figure 18. Binding energies between graphene/N-doped graphene and zinc.

Comment #4: *The authors stated "enlarged specific surface area" as a weakness for 3D matrices, but went on to synthesize carbon matrices with high surface area anyway, and these matrices somehow do not come with any of the perceived disadvantages of 3D matrices such as aggravated corrosion of zinc and HER.*

Response: We are very sorry for our inaccurate statement. We have made corresponding modifications in the manuscript, and the details of discussion are as follows:

Compared with the 2D planar current collector, the specific surface area of the 3D current collector is larger, and the local current density of the 3D current collector is lower than that of the planar current collector, which can be proved by Sand's time model (J Am Chem Soc 2015, 137, 15209).

$$\tau_s = \pi D \left(\frac{e z_c c_0}{2J} \right)^2 \left(\frac{\mu_a + \mu_c}{\mu_a} \right)^2 \quad (1)$$

where e is the elementary charge, μ_a and μ_c are the anionic and the cationic mobilities, respectively, D the ambipolar diffusion constant $D = (\mu_a D_c + \mu_c D_a) / (\mu_c + \mu_a)$, D_a and D_c being the anionic and cationic diffusion constants, respectively, J is the current density, C_{c0} the initial cationic concentration and z_c is the cationic charge number. In the case where the anionic and cationic charge numbers z_a and z_c are equal $(\mu_a + \mu_c) / \mu_a = 1/t_a$, where t_a is the anionic transport number ($t_a = z_a \mu_a / (z_a \mu_a + z_c \mu_c)$). It can be seen from formula (1) that τ_s is directly proportional to the square of the initial concentration (c_0)

and inversely proportional to the square of the current density (J). This shows that dendrite formation is easier under high current density (J), and the time of dendrite formation can be prolonged by reducing current density. 3D hosts with the enlarged specific surface area can play a key role in reducing the local current density, uniform electric field distribution and ion concentration gradient.

When conventional porous carbon with pores ranging from tens to hundreds of microns is used as a 3D current collector for metal anode, the pore structure with a large size is not large enough to effectively disperse the surface current due to the small specific surface area. Several issues need to be resolved, however, for the practical applications of such scaffolds. For example, nanostructures in anodes induce a high electrode/electrolyte interface area that causes significant consumption of electrolyte and metal (such as Zn, Li, Na) species, and consequently low coulombic efficiency because of the formation of an excessive amount of SEI. Therefore, the pore size distribution and surface area of 3D hosts can be adjusted to optimize the metal deposition and electrochemical performance.

In this stage, we revised the SSA data of different carbon matrices to acquire a better understanding. As shown in **Figure 19**, the 3D-LFGC and 3D-RFGC matrices possessed a higher SSA of 181.06 and 97.13 $\text{m}^2 \text{g}^{-1}$, respectively, compared with the pristine 3D-LC (40.21 $\text{m}^2 \text{g}^{-1}$) and 3D-RC (61.0 $\text{m}^2 \text{g}^{-1}$). The growth of VGs and GFs provided enlarged specific surface area and active spaces, which introduces the abundant zincophilic sites (N-doped graphene) and induces a homogeneous current density (mutiphysical field simulation). The 3D-LFGC@Zn and 3D-RFGC@Zn have an SSA of 37.85 and 16.12 $\text{m}^2 \text{g}^{-1}$, respectively. The 3D-RFGC@Zn indicates a relatively low SSA, demonstrating a dense 3D composite Zn anode. The 3D-LFGC@Zn with larger SSA maybe suffer from serious Zn corrosion and the hydrogen evolution reaction (HER) after repeated cycles.

Figure 19. The comparison of specific surface area of different carbon matrices.

In this manuscript, interfacial side reactions can be well suppressed as shown in **Figure 20**. Zn corrosion is analyzed by linear polarization measurements in 2 M ZnSO₄ electrolyte, where the corrosion potential of 3D-RFGC@Zn (-0.958 V), 3D-LFGC@Zn (-0.983 V) was higher than the bare Zn electrode (-1.014 V). Moreover, after depositing/stripping certain cycles in ZnSO₄ electrolyte, massive by-products, Zn₄SO₄(OH)₆ · 5H₂O were generated due to severe corrosion for bare Zn electrode, whereas there are nearly no obvious by-products on 3D-RFGC@Zn electrode according to the results of ex-situ XRD patterns (**Figure 20b**), which indicates that the 3D hierarchical graphene matrix can protect Zn anode from corrosion. From **Figures 20c-20d**, a great mass of macroscopic passive aggregates was continuously forming and detaching from the Zn surface. In sharp contrast, the 3D-RFGC surface maintained a smooth morphology with no clutter observed. The reason is that the 3D graphene host is helpful to the uniform deposition of Zn metal, which makes it possible to obtain a dense and stable electrolyte/Zn interface, thus slowing down the Zn corrosion and other side reactions, which is also demonstrated by the XPS depth profiles after the cycles.

Figure 20. (a) Linear polarization curves showing the corrosion on bare Zn and 3D-Zn. (b) XRD patterns of different Zn anodes after 10 cycles and 50 cycles. Three-electrode

system of (c) bare Zn plates and (d) 3D-RFDC host in transparent tanks representing the side reactions visually during continuous Zn plating/stripping at a current density of 5 mA cm².

Comment #5: *The cartoons in Figure 1 and 2 are missing legend.*

Response: We have included the legends in Figure 1 and Figure 2 in this revision, and the possible errors in the manuscript have been revised.

Comment #6: *Figure 5a-e are missing scale bars.*

Response: We have included the scale bars in Figures 5a-5e in this revision, and the possible errors in the manuscript have been revised.

Comment #7: *The authors claimed a preference for zinc ions to deposit on the top of a control sample with Figure S19 and Video S2, but neither display item shows vertical information. The items that do show vertical information for the same sample, Figure S21 and S22, do not show any difference between the top and the bottom.*

Response: To enhance the accuracy of the model, we re-checked the model of 3D-LC, 3D-RC, 3D-LFGC and 3D-RFGC, and showed vertical information of the simulated Zn²⁺ ion concentrations and current distributions. The results and discussions are as follows:

According to the COMSOL simulations, we established models to prove the action of multi-function layer of GFs and VGs in regulating current density and Zn-ion flux distributions, as depicted in **Figure 21**. The Zn-ion flux is indeed more concentrated on the 3D-LC, bringing about ever-growing dendrites with continuous consumption of the Zn electrode and electrolyte. With the deposition time increasing to a certain extent, evident destruction with uncontrolled dendrites would form on the electrode surface (**Figure 22d-22f**). In the SEM evidence, Zn metal deposited in a large amount near the top of the 3D-LC and blocked the vertically-oriented carbon channel. The deposition in the carbon channel was further hindered, which greatly slowed down the deposition rate and the utilization rate of the 3D-LC. Fortunately, after fabricating the layer of GFs and VGs, the Zn²⁺ flux in this architecture with abundant zincophilic sites is effectively homogenized with bottom-up deposition. The whole electrode surface gets smooth with uniform and compact Zn deposition while the whole empty space is gradually filled as

the deposition time increases (**Figure 22a-22c**). For the 3D-RC and 3D-RFGC, The top of 3D-RC showed lower zinc ion concentration distribution than other areas, thus leading to uneven Zn deposition and dendrites growth with “tip effect” (**Figure 23d-23f**). The results of 3D-RFGC indicate uniform Zn ions distribution and high reaction rate in the channels, which proves that zinc metal is preferentially deposited in the channels because of the graphene layers.

Figure 21. The geometrical structures and boundaries condition for (a) 3D-LC; (b) 3D-LFGC; (c) 3D-RC and (d) 3D-RFGC.

Figure 22. Simulation results of the Zn^{2+} ions concentration distribution of 3D-LFGC and 3D-LC at a, d) 0 s; b, e) 60 s; c, f) 120 s.

Figure 23. Simulation results of the Zn^{2+} ions concentration distribution of 3D-RFGC and 3D-RC at a, d) 0 s; b, e) 60 s; c, f) 120 s.

As shown in **Figure 24**, the introduction of 3D conductive GFs and VGs matrices with increased surface area enables the effective lowering of current density. This is superior to the scenario of the pristine 3D-LC (**Figures 24d-24f**). According to Sand's time model, lowering the local current density would delay the initial formation point of the dendrite. For 3D-RFGC and 3D-RC, 3D-RFGC matrices could afford a uniformly distributed electric field, where Zn can be evenly deposited onto the skeletons of GFs and VGs due to the relatively small current gradient created at the early stage as well as maintained during the subsequent cycling (**Figures 25a-25c**). In contrast, a 3D-RC without the graphene skeletons tends to initiate “tip-nucleation” and further evolves into uneven current distributions, as illustrated in **Figures 25d-25f**. In this respect, Zn dendrites would emerge sequentially. In these models, a constant current is applied, so the current density does not change with the reaction time, which is determined by the initial state of the materials.

Figure 24. Simulation results of the current density distribution of 3D-LFGC and 3D-LC at a, d) 0 s; b, e) 60 s; c, f) 120 s.

Figure 25. Simulation results of the current density distribution of 3D-RFGC and 3D-RC at a, d) 0; b, e) 60 s; c, f) 120 s.

Responses to the comments by Reviewer #2

Reviewer #1: Here, Mu et al. report a 3D hierarchical graphene matrix layer deposited on the Zn anode as an efficient tool for the longitudinal deposition and an inhibitor for the zinc dendrite growth, leading to extended cycle life for the half and full cells. Overall, the prolonged cycle data and calculated simulation data are appealing but there seem to be some major fundamental inconsistencies that do not explain the kind of performance claimed and communicated here. Experimental details are also not obvious, making it difficult to interpret the results.

Comment #1: *In the main text Page 8, line 2-5, the graphene has zincophilicity and enough space to store Zn. Then, what is the specific evidence or reference? There is BET data that the 3D-FGC is estimated to be 142 m²/g but it seems difficult to support the fact that it reduces the concentration gradient of Zn²⁺ ions and enables uniform impregnation of zinc ions. (pg 8, line 15-20).*

Response: Thank you for your constructive comments. We have added and explained the zincophilicity of graphene and the concentration gradient of Zn²⁺ ions distribution, as follows:

(1) The zincophilicity of N-VGs and GFs clusters

First, ammonia (NH₃) treatment at 1050°C followed the growth of VGs and GFs, which resulted in nitrogen (N) doping, as demonstrated by XPS data (**Figure 1**). The introduction of VGs and GFs provided enlarged specific surface area and active spaces (**Figure 2**), which provides abundant zincophilic sites (N-doped graphene), consistent with previous result reported in the reference (Adv Mater 2020, 32, e2003425).

Figure 1. a) The device of ammonia (NH₃) treatment for 3D hosts with VGs and GFs.

b) XPS spectrum of N-doping for 3D-RFGC.

Figure 2. N₂ adsorption–desorption isotherms of a) 3D-LC, b) 3D-RC, c) 3D-LFGC, d) 3D-RFGC, e) 3D-LFGC@Zn, f) 3D-RFGC@Zn, and g) the comparison of various SSA data.

Second, we have supplemented **contact angle measurements** of the different carbon matrices prepared in this manuscript, which are used in combination to account for enhanced zincophilicity. Attributed to the abundant N-doped sites on the surface of GFs and VGs skeletons, the contact angles refer to the 2 M ZnSO₄ electrolyte significantly decreases to 7.92° for 3D-RGFC (**Figure 3**), implying an enhanced electrolyte permeability. This result ensures that even at high Zn deposition and high cathode loadings, the electrolyte can remain fully infiltrated, thus maintaining an

efficient ion transport process. Specifically, we performed contact angle tests on bare Zn, 3D-LC, 3D-RC, 3D-LFGC, 3D-RFGC, 3D-LFGC@Zn, and 3D-RFGC@Zn, respectively. As shown in **Table 1**, we concluded that the N-doped graphene matrices exhibited excellent wettability of electrolyte.

Figure 3. Optical images of contact angles on various current collectors in this work.

Table 1. The contact angle of various current collectors in this work.

Samples	1s	5s	10s	20s
Bare Zn	51.96	43.63	40.35	40.35
3D-LC	134.10	133.50	133.50	133.50
3D-RC	121.76	120.30	120.10	120.10
3D-LFGC	54.31	39.57	28.28	22.92
3D-RFGC	29.51	14.61	9.68	7.92

Third, **density functional theory (DFT) theoretical simulation** was also used to account for enhanced zincophilicity. The experimental details and the discussion are summarized as follows:

The introduction of zincophilic sites is critical in adjusting the absorbing/bonding ability of 3D hosts (*Adv Mater* 2020, 32, e2003425; *Adv Mater* 2021, 33, e2101649). We, therefore, conducted DFT computations to compare zincophilic ability of nitrogen and carbon in graphene models (**Figure 4**). The configurations of zinc adsorption on different sites on carbon substrates are illustrated in **Figure 5**. As is shown in **Figure 6**, the binding energies of zinc-atom on different adsorption sites of graphene are, -0.0196

(Top), -0.0203 (Hollow), and -0.0180 (Bridge) eV, respectively. For the adsorption sites on N-doped graphene, the corresponding binding energies are, -0.0246 (Top), -0.0270 (Hollow), and -0.0254 (Bridge) eV, respectively. This finding underscores that the introduction of nitrogen-doping changes the interaction of zinc and carbon substrate from zincophobic to zincophilic. Therefore, the 3D-FGC matrices with zincophilic sites exhibited a uniform Zn deposition, together with improved electrochemical performances.

Figure 4. Configurations of a) pristine graphene and b) nitrogen-doped graphene.

Figure 5. Configurations of zinc adsorption on different sites of graphene/nitrogen-doped graphene.

Figure 6. Binding energies between graphene/N-doped graphene and zinc.

(2) The concentration gradient of Zn^{2+} ions

For one thing, we updated the BET data of different 3D carbon matrices to acquire a better understanding. As shown in **Figure 2** in comments#1, the 3D-LFGC and 3D-RFGC matrices possessed a higher SSA of 181.06 and 97.13 $m^2 g^{-1}$, respectively, compared with the raw 3D-LC (61 $m^2 g^{-1}$) and 3D-RC (40.21 $m^2 g^{-1}$). The growth of VGs and GFs provided enlarged specific surface area and active spaces, which introduced the abundant zincophilic sites (N-doped graphene) and induced a homogeneous current density. This conclusion was also mentioned in many previous reports (Adv Mater 35, e2205206 (2023); Advanced Functional Materials 31, (2021); Advanced Functional Materials 31, (2021)). Furthermore, from **Figures 2e-2g**, the 3D-LFGC@Zn and 3D-RFGC@Zn have an SSA of 37.85 and 16.12 $m^2 g^{-1}$, respectively. This result indicates a relatively low SSA for 3D-RFGC@Zn anode, demonstrating a dense 3D composite Zn anode. Therefore, a 3D host with a large SSA can regulate the uniform plating and stripping of Zn.

For another, we further simulated the zinc ion concentration and current density distribution using COMSOL Multiphysics (Supporting Information). The geometrical structures and boundaries condition for 3D-LC, 3D-RC, 3D-LFGC and 3D-RFGC are shown in **Figure 7**. The current density i and reaction time are 100 mA/cm^2 and 120 s assumed in these models.

Figure 7. The geometrical structures and boundaries condition for (a) 3D-LC; (b) 3D-LFGC; (c) 3D-RC and (d) 3D-RFGC.

It is shown that the local current density on the top surface of the pristine 3D-LC is larger than that of the 3D-LFGC matrices (**Figure 8**). The high local current density is forced to grow Zn dendrites on the upside of the 3D-LC. The introduction of 3D conductive GFs and VGs matrices with increased surface area enables the effective lowering of current density. This is superior to the scenario of the pristine 3D-LC (**Figures 8d-8f**). According to Sand's time model, lowering the local current density would delay the initial formation point of the dendrite.

For 3D-RFGC and 3D-RC, 3D-RFGC matrices could afford a uniformly distributed electric field, where Zn can be evenly deposited onto the skeletons of GFs and VGs due to the relatively small current gradient created at the early stage as well as maintained during the subsequent cycling (**Figures 9a-9c**). In contrast, 3D-RC suffers from a higher current density distribution than 3D-RFGC does, which leads to a higher consumption of zinc ions in the region, resulting in more zinc metal deposition. These results indicate uneven zinc deposition and dendrite growth in subsequent stages.

Figure 8. Simulation results of the current density distribution of 3D-LFGC and 3D-LC at a, d) 0 s; b, e) 60 s; c, f) 120 s, respectively.

Figure 9. Simulation results of the current density distribution of 3D-RFGC and 3D-RC at a, d) 0 s; b, e) 60 s; c, f) 120 s, respectively.

The simulated distributions of Zn^{2+} ion concentration indicates that Zn^{2+} ion flux is indeed more concentrated on the 3D-LC, bringing about ever-growing dendrites with continuous consumption of the Zn electrode and electrolyte. With the deposition time increasing to a certain extent, evident destruction with uncontrolled dendrites would form on the electrode surface (**Figures 10d-10f**). Fortunately, after fabricating the layer of GFs and VGs, the Zn^{2+} flux in this architecture with abundant zincophilic sites is effectively homogenized with bottom-up deposition. The whole electrode surface gets smooth with uniform and compact Zn deposition and the whole empty space is gradually filled as the deposition time increases (**Figures 10a-10c**). For the 3D-RC and 3D-RFGC, The top of 3D-RC showed lower zinc ion concentration distribution than other areas, thus leading to uneven Zn deposition and dendrites growth with “tip effect” (**Figures 11d-11f**). The results of 3D-RFGC indicate a uniform distribution of Zn ion concentration and a high reaction rate in the channels, which proves that zinc metal is preferentially deposited in the channels because of the graphene layers (**Figures 11a-11c**).

Figure 10. Simulation results of the Zn^{2+} ions concentration distribution of 3D-LFGC and 3D-LC at a, d) 0 s; b, e) 60 s; c, f) 120 s.

Figure 11. Simulation results of the Zn^{2+} ions concentration distribution of 3D-RFGC and 3D-RC at a, d) 0 s; b, e) 60 s; c, f) 120 s.

Comment #2: *In the Figure 4a, the symmetric cell data of 3D-RFGC at the current density 1 mA/cm^2 does not appear to be effective compared to other papers (Ref. Nat. Commun, 13, 7922 (2022) & Nat. Commun. 11, 3961 (2020)). The author should have to load another data that shows constant overpotential for the entire cycling. If not, the sudden increase in overpotential, which occurs even in the best data (about 2000 h \sim 2500 h in the 3D-RFGC and about 2300 h \sim 4500 h in the 3D-LFGC), would require further proof which can prove that it is not a short circuit.*

Response: Based on your comments, we have further double-checked the Zn/Zn symmetrical cells data in this manuscript and the literature you mentioned, and we have included the voltage curves at different durations and voltage hysteresis of the entire cycling to show the polarization in symmetric cells. As shown in **Figure 12**, although there is a voltage fluctuation, the enlarged curves of the depositing/stripping curves are still normal in the whole process of testing (**Figures 12b-12e**), which also exists in other reports⁶⁻⁸. The voltage hysteresis of the entire cycling also indicates large voltage fluctuations in two areas, and gradually resumes the normal deposition and stripping

process (**Figure 12f**). This phenomenon still exists under higher current densities and capacities. However, we can still confirm that it is not a short circuit through the enlarged deposition/stripping curves.

Figure 12. The enlarged voltage curves (a-e) at different stages and (f) hysteresis of the entire cycling to show the polarization in symmetric cells.

Comment #3: *There is excellent paper (Ref. Electrochimica Acta, 2022, 425, 140648) about coating Zr-MOF on Zn metal anode. The authors should make note of them.*

Response: We have cited this work as *Ref. 9* to enrich our research background in the revised manuscript.

Comment #4: *The electrochemical performance of different electrodes of bare Zn, 3D-LFGC matrix, and 3D-RFGC matrix are shown in the figure 3a-i. I don't know if bare Zn is an appropriate comparison. For example, there is a reference study that has achieved good results by transforming zinc metal into a 3D matrix structure (Ref. Adv. Energy Mater.2022, 12, 2102797). Rather than looking at the morphology of plating in bare Zn, which is a 2D structure, it is considered that a more effective comparison method is to look at the plating process by processing zinc metal like the 3D structure presented in this study. Also, in figure 3j, there is not any information about the cell you used in this overpotential data. It would be nice to additionally fill in the information of the cell used in this graph.*

Response: According to your suggestion, we have made the following modifications and discussions to address these issues in the revised manuscript:

(1) The issue of comparison standards:

We have consulted some literature about zinc metal anodes. In fact, the performance comparison standards of symmetrical cells are different, and there are some works using 2D zinc anode to compare with the various 3D zinc anode (*Energy Storage Mater. 30 (2020) 104-112; Adv. Funct. Mater. 2023, 33, 2208288; ACS Nano 2021, 15, 15259-15273; Adv. Mater. 2022, 34, 2110047*). Of course, we agreed more with your view of comparing performance on a 3D scale; therefore, we purchased a commercial foam Zn directly and conducted short-term symmetric cells tests, and we also added the data of symmetric cells for 3D Zn foam in the revised manuscript. Details as follows:

SEM images show that the porous structure of foam Zn, like foam nickel and foam copper, has a large pore structure of about 10 μm . The magnified SEM images show that zinc metal exists in the form of flake and powder, and its surface is relatively rough, with a large number of holes. The thickness is about 400 μm . It is appropriate to use this as the benchmark for the experiment (**Figure 13**).

Figure 13. SEM images of commercial foam Zn from the top-view and cross-view.

We tested the performance of symmetrical cells of Zn foam/Zn foam. As shown in **Figure 14a**, the 3D foam zinc cell has a small voltage at $1 \text{ mA cm}^{-2}/1 \text{ mAh cm}^{-2}$. With the increase of current density and capacity, the voltage curve fluctuates greatly under the current of 10 mA cm^{-2} . The zinc metal can still be deposited under the current of 80 mA cm^{-2} , but the voltage fluctuation is large, and the cell is short-circuit after a period of cycling (**Figure 14**).

Figure 14. Symmetrical cells of 3D Zn foam operating at a) 1 mA cm^{-2} and 1 mAh cm^{-2} ; b) 10 mA cm^{-2} and 10 mAh cm^{-2} ; c) 20 mA cm^{-2} and 20 mAh cm^{-2} ; d) 80 mA cm^{-2} and 80 mAh cm^{-2} .

(2) For *Figure 3j*, we have made adjustments to the Figures, including the over-potential values and the corresponding materials as well as experimental conditions, to

make the data comparisons clearer. In the revised manuscript, *Figures 3k, 4l, 4k, 4l, Tables S3-S4* have been modified according to your comments.

Comment #5: *In the figure 2p, it seems that contact angle measurement has been conducted, but the electrolyte used during the experiment is not mentioned in the paper or SI. It would be nice to add information about electrolytes, Looking at figure 2p, it seems that the contact angle analysis was conducted, but the electrolyte used during the experiment is not mentioned in the text or SI. It would be nice to add information about electrolytes to the graph or to the text. And it is stated that the wettability is increased due to uniform N doping, and it would be much better to understand if you explain in detail the evidence reference or mechanism that supports this.*

Response: According to your comments, we have added the experimental details of contact angle (CA) measurements in *SI*. The electrolyte used during the CA experiment is 2 M ZnSO₄. Furthermore, we conducted CA measurements and density functional theory (DFT) theoretical simulations for the different carbon matrices prepared in this manuscript, to support our view of the enhanced wettability. The experimental details and the discussion are summarized as follows:

(1) Contact angle measurements

Attributed to the abundant N-doped sites on the surface of GFs and VGs skeletons, the contact angle refers to the 2 M ZnSO₄ electrolyte significantly decreases to 7.92° for 3D-RGFC (**Figure 3**), implying the enhanced electrolyte permeability. This ensures that even at high Zn deposition and high cathode loading the electrolyte can remain fully infiltrated, thus maintaining an efficient ion transport process. Specifically, we performed contact angle tests on bare Zn, 3D-LC, 3D-RC, 3D-LFGC, 3D-RFGC, 3D-LFGC@Zn, and 3D-RFGC@Zn, respectively, and recorded the process within 20 s. As shown in **Table 1** below, we concluded that the N-doped graphene matrices exhibit excellent wettability.

(2) Density functional theory (DFT) theoretical simulations

The introduction of zincophilic sites is critical in adjusting the absorbing/bonding ability of 3D hosts (*Adv Mater* 32, e2003425 (2020); *Adv Mater* 33, e2101649 (2021)). We, therefore, conducted DFT computations to compare zincophilic ability of nitrogen

and carbon in graphene models (**Figure 4a**). The configurations of zinc adsorption on different sites on carbon substrates are illustrated in **4b**. As is shown in **4c**, the binding energies of zinc-atom on different adsorption sites on graphene are, , -0.0196 (Top), -0.0203 (Hollow), and -0.0180 (Bridge) eV, respectively. For the adsorption sites on N-doped graphene, the corresponding binding energies are, -0.0246 (Top), -0.0270 (Hollow), and -0.0254 (Bridge) eV, respectively. This finding underscores that the introduction of nitrogen-doping changes the interaction of zinc and carbon substrate from zincophobic to zincophilic. Therefore, the 3D-FGC matrices with zincophilic sites exhibited a homogenous Zn deposition, together with boosted electrochemical performances.

Comment #6: *In the main text Page 14, line 20-24, and figure 5j, you identified the new peak as the $Zn_4SO_4(OH)_6 \cdot 3H_2O$. Do you have any references to support this?*

Response: We re-checked the XRD data of different Zn anodes after 10 cycles and 50 cycles. Note that it is challenging to exactly distinguish the intermediate products by XRD. New phases such as $Zn_4SO_4(OH)_6 \cdot 0.5H_2O$ (JCPDS#44-0674), $Zn_4SO_4(OH)_6 \cdot H_2O$ (JCPDS#39-0690), $Zn_4SO_4(OH)_6 \cdot 3H_2O$ (JCPDS#39-0689), $Zn_4SO_4(OH)_6 \cdot 4H_2O$ (JCPDS#44-0673) and $Zn_4SO_4(OH)_6 \cdot 5H_2O$ (JCPDS#39-0688) are observed (**Figure 15**). Relevant literature has been cited in the revised manuscript to support the complex by-products of $Zn_4SO_4(OH)_6 \cdot xH_2O$ (*Adv. Funct. Mater.* 2022, 32, 2207732; *Adv. Energy Mater.* 2021, 11, 2101158; *ACS Energy Lett.* 2019, 4, 2776-2781).

Figure 15. XRD patterns of a) different Zn anodes after 10 cycles and 50 cycles and b) the standard card of by-products.

Comment #7: *In the Supporting Information Page 2, materials section, it was written that ZnSO₄ (2M) and ZnSO₄/MnSO₄ (2M+0.5M) were used when assembling the coin cell. However, in the Supporting Information Page 5, in situ optical microscope characterization and electrochemical performance assessment sections, it was written that 2M MnSO₄ was used as the electrolyte. Please check which information is correct.*

Response: We used an electrolyte of 2 M ZnSO₄ in the in-situ optical microscope characterization and electrochemical performance assessment. Specifically, 2 M electrolyte was used in the coulombic efficiency (CE) test, the symmetric cells test, the in situ observation test, the various Zn/V₂O₅ full cells and zinc ion capacitor electrochemical tests, only ZnSO₄/MnSO₄ (2 M+0.5 M) was used in the Zn/MnO₂ full cells test, which could prevent the dissolution of Mn²⁺ from Mn³⁺ disproportionation and reduce the accumulation of the byproducts from both cathode and anode.

Responses to the comments by Reviewer #3

Reviewer #1: The authors developed a 3D hierarchical graphene matrix consisting of wood-derived multichannel carbon, N-doped graphene nanofibers clusters, and vertical graphene arrays. The matrix worked well when serving as a substrate for Zn metal anodes. The electrochemical performances are excellent. However, some flaws and concerns also need to be addressed.

Comment #1: *I suggest the authors add more schematics and text descriptions of 3D-LFGC and 3D-RFGC. For example, how were 3D-LFGC and 3D-RFGC cut from the original 3D-FGC? What are the characteristic morphologies of the 3D-LFGC and 3D-RFGC electrodes?*

Response: Thank you for your constructive comments. We have added additional experiments and discussions to further describe the preparation process and structural characterization of the multichannel graphene matrices mentioned in the revised manuscript and SI, as follows:

1) Synthesis of Graphene Arrays

In this manuscript, we for the first time develop a 3D hierarchical graphene matrix consisting of nitrogen-doped graphene nanofibers clusters (GFs) anchored upon vertical graphene arrays (VGs) modified multichannel carbon (3D-FGC). The schematic illustration of fabricating the 3D-FGC matrices is shown in **Figure 1**.

Figure 1. The schematic illustration of the fabrication of the 3D-LFGC and 3D-LFGC matrices.

As shown in **Figure 1**, the overall synthesis process of 3D-FGC matrices consists

of three steps: i) carbonization to obtain porous carbon channels; ii) CVD growth of GFs and VGs and iii) post-treatment of ammonia (NH₃) atmosphere.

i) Synthesis of Multichannel Carbon Matrix

First, the natural basswood (Jiangxi Three-Wood Technology Co., Ltd.) was cut along the radial direction and longitudinal direction (**Figure 2**). Wood can be cut into different sizes, shapes and thicknesses (**Figures 3a-3c, and 4a-4c**). In this work, square or round wood slices with a thickness of 0.3~0.8 mm were used (**Figure 3d**). Second, the wood slices were stabilized in the air at 260 °C for 6 h. Third, the carbonization process was conducted at 1100 °C for 6 h in ammonia (NH₃, 50 mL min⁻¹). The carbonized wood was activated in a carbon dioxide (CO₂, 100 mL min⁻¹) flow at 750 °C for 12 h to obtain the activated multichannel carbon matrix (**Figure 3e**). The carbon matrices shown in **Figure 3** were named 3D-LC. The annealed carbon matrices can be polished with 2500 grit sandpapers, and cleaned by ultrasonic washing with deionized water and ethanol, thus obtaining low-tortuosity carbon matrices. The 3D-LC with different thicknesses from 120 μm to 800 μm can be obtained shown in **Figures 4a-4f**.

Figure 2. Schematic diagram of the natural basswood processing.

Figure 3. The optical photographs of raw wood and after different treatments including a) wood from the tree; b) different sizes and shapes; c) different thicknesses; d) used in this work and e-g) different size wood-derived multichannel carbon (3D-LC).

Figure 4. The 3D-LC matrices with different thicknesses of a) 800 μm , b) 420 μm , c) 360 μm , d) 280 μm , e) 160 μm and f) 120 μm .

Similarly, the carbon matrix cut along the radial direction (named 3D-RC) was obtained by the same process except for the first step of the cutting process, shown in **Figure 5**. Also, different sizes, shapes and thicknesses of wood slices can be manufactured in this work (**Figure 6**). During the carbonization process, the thickness and size of the wood chips were shrunk.

Figure 5. The optical photographs of raw wood and after different treatments including a) wood from the tree; b) cross-sectional image; c) different sizes and shapes; d) pre-oxidation and e) carbonized multichannel carbon (3D-RC).

Figure 6. The 3D-RC matrices with different thicknesses of a) 800 μm , b) 550 μm , c) 450 μm , and d) 300 μm .

ii) CVD growth of GFs and VGs and N-doping treatment

In this stage, we successfully grew GFs and VGs on the two carbon substrates obtained above by thermal chemical vapor deposition (thermal-CVD) technology. By controlling the growth temperature, atmosphere and time, a complete growth process was established for the first time. In particular, in this manuscript, we report the first finding that thermal-CVD technology can achieve synchronous growth of GFs and VGs by prolonging growth time and increasing the proportion of carbon sources (CH_4). The experimental process is shown below (**Figure 7**):

Figure 7. The device of growing GFs and VGs by a thermal-CVD method.

At 1050 °C, the methane (CH_4 , 20-60 mL min^{-1}) used as a carbon source was introduced into the quartz furnace when the temperature reached 1050 °C. The GFs and VGs can be grown successfully between 1050 °C on the pristine carbon matrices by introducing H_2 (80-120 mL min^{-1}). The concentrations of H_2 and CH_4 varied with the temperature. The VGs and GFs heights can be controlled by varying the growth time. The as-obtained graphene productions can be named 3D-LFGC and 3D-RFGC, respectively. After finishing the growth, turn off the methane and hydrogen, respectively, the introduction of ammonia, and keep 30 minutes at 1050 °C, so as to achieve nitrogen doping.

SEM images, TEM images, XRD patterns, Raman spectra, TGA curves, specific surface area and micro-CT were conducted to analyze the structure and composition of 3D-LC, 3D-RC, 3D-LFGC and 3D-RFGC. In the revised manuscript, we made a detailed supplement and analysis of the two carbon materials, as seen in the main text of the manuscript and SI.

Figure 8. SEM images of 3D-LC and 3D-RC.

Figure 8 indicates the basic characteristic of the multichannel carbon matrices for 3D-LC and 3D-RC from the SEM images; the micro-CT images also show the porous and continuous structure, which is beneficial to deposit Zn metal and loading cathode

materials of AZIBs.

The morphologies of 3D-LFGC and 3D-RFGC are shown in **Figure 9** and **Figure 10**. At this stage, we regulated the concentration of the reaction atmosphere (Ar/CH₄/H₂) to find the best growth conditions and extended the growth time, thus obtaining VGs and GFs with different heights and densities. The VGs are firmly anchored on the inner walls of the carbon channel, while the GFs are grown vertically from the surface of the carbon matrix directly and are shrouded in the upper layer of the VGs. We have investigated the origin of the GFs. Based on previous studies¹⁻³, we compared the growth of vertical graphene on the surface of pure carbon-based materials such as carbon black particles, carbon fiber (CF) and carbon nanofiber (CNF), and found that only VGs exist on the surface of the above-mentioned matrix, and there are no GFs (**Figure 11**). In this regard, EDX mapping analysis of the original carbonized wood revealed the presence of a variety of micronutrients in the original wood (**Figure 12**), which may have contributed to the catalytic growth of carbon nanofibers, as reported in the literature^{4, 5}. Further, by verifying catalytic effect by immersing the wood in an aqueous solution of ferric nitrate (Fe(NO₃)₃·9H₂O), we found that the nanofibers clusters grew more rapidly in the presence of the catalyst (**Figure 13**), this further verifies our hypothesis of catalytic effect (*Adv. Energy Mater.* 2011, 1, 1205-121; *Energy Environ. Sci.*, 2010, 3, 1286-1293). This provides ideas for the rapid preparation of gaseous carbon nanofiber composite materials.

Figure 9. SEM images of 3D-LFGC host for longitudinal direction when extended the growth time a) 2 h; b) 6 h; c) 9 h and d) 12 h.

Figure 10. SEM images of 3D-RFGC host for radial direction when extended the growth time a) 2 h; b) 6 h; c) 9 h and d) 12 h.

Figure 11. SEM images of a) carbon black, b) carbon nanofibers and c) carbon fibers.

Figure 12. EDX mapping of carbonized wood including a) SEM image, b) C, c) N, d) O, e) Mg, f) Ca, g) Fe, h) Zn, i) Na. (The scale bar is 20 μm)

Figure 13. SEM image of 3D-RFGC in the presence of catalyst.

Therefore, based on your suggestion, we make further supplements to the structural and composition characterizations of the 3D-RFGC and 3D-LFGC in the first part of the revised manuscript and supporting information.

Comment #2: *The details of the coin cell assembly are not clear. What are the final dimensions of the 3D-LFGC and 3D-RFGC electrodes? What is the electrolyte volume? How did the authors ensure the electrolyte wetting of the entire thick 3D electrode?*

Response: We have added experimental details to the revised supporting information, including cell assembly, electrode sizes, electrolyte volume, and electrode pre-

treatment, as follows:

(1) The Zn half-cells and full-cells were assembled in CR2032 coin cells. In symmetrical cells tests, the bare Zn foil is a small circle with a diameter of 10 mm, and the sizes of 3D-RC, 3D-LC, 3D-RFGC and 3D-LFGC are about $5 \times 5 \times 0.3$ mm, $5 \times 5 \times 0.5$ mm, $5 \times 5 \times 0.3$ mm and $5 \times 5 \times 0.5$ mm (length \times width \times height), respectively. In full cells, the cathode materials were loaded on the 3D-host without GFs by vacuum filtration. The dimensions of the 3D-LC for cathodes in full cells is about $5 \times 5 \times 0.5$ (length \times width \times height), and the dimensions of the 3D matrices for anode in full cells is about $6 \times 6 \times 0.3$ (length \times width \times height).

(2) The electrolyte volume of 2 M ZnSO_4 is ~ 75 μl in the CE and symmetrical cells tests, according to the coin-cell protocol (*Nature Energy* 2020, 5, 561-568). In this paper mentioned, the suggested coin cell parameters, an electrolyte amount (75 μl), are valid for all symmetric metal cells, for example Mg/Mg, Zn/Zn cells.

(3) In order to ensure sufficient wetting of the electrolyte, the electrodes are usually immersed in 2 M ZnSO_4 electrolyte for 5~10 s before cell assembling, and the 3D-RFGC and 3D-LFGC matrices we developed have abundant N-doped sites, excellent electrolyte wettability can be directly seen from the contact angle data (Figure 14, Table 1).

Figure 14. Optical images of contact angles on various current collectors in this work.

Table 1. The contact angle of bare Zn and various matrices used in this work.

Samples	1 s	5 s	10 s	20 s
Bare Zn	51.96	43.63	40.35	40.35
3D-LC	134.10	133.50	133.50	133.50
3D-RC	121.76	120.30	120.10	120.10
3D-LFGC	54.31	39.57	28.28	22.92
3D-RFGC	29.51	14.61	9.68	7.92

Comment #3: *What are the electrochemically active surface areas of 3D-LFGC and 3D-RFGC?*

Response: We have added experiments to reveal the electrochemically active surface area (ECSA), the procedure and discussions of the experiments are as follows:

(1) Experiments details:

The ECSAs of 3D-LC, 3D-RC, 3D-LFGC and 3D-RFGC electrodes were estimated according to an established non-aqueous electrochemical double-layer capacitance (DLC) method (*Nano Energy 2020, 75; Energy Environ. Mater. 2022, 0, 1-7*). The DLC measurement was conducted in the aqueous electrolyte of 2 M ZnSO₄. A three-electrode setup was employed in the CV testing with a 3D-LFGC or 3D-RFGC working electrode, a Pt wire counter electrode, and a reference electrode of Ag/AgCl. For each DLC measurement, CV curves were acquired in a narrow potential window of ± 50 mV of the open circuit potential (OCP) at scan rates of 5, 10, 20, 40, 60, 80 and 100 mV s⁻¹. The capacitive current can be calculated from the difference between the anodic and cathodic current densities ($\Delta j = j_a - j_c$) at the OCP in the collected CV curves. The ECSA can be determined by the following equation: $ECSA = C_{dl}/C_s$, where C_s is the specific capacitance of the sample of an atomically smooth planar surface of the material per unit area under identical electrolyte conditions. For our estimation of the ECSA, general specific capacitances of $C_s = 40 \mu\text{F cm}^{-2}$ based on typical reported values is used (*Energy Environ. Mater. 2022, 0, 1-7*).

(2) Results and discussions

Figure 15 shows that the double-layer capacitance (C_{dl}) of pristine 3D-LC electrode is much lower (1.04 mF cm^{-2}) than that of the pristine 3D-RC electrode (2.5 mF cm^{-2})

(**Figures 15a-15b**). However, after the growth GFs and VGs, the Cdl of the formed 3D-RFGC electrode is significantly increased to 15.7 mF cm² (**Figure 15e**), which is higher than that of 3D-LFGC (12.51 mF cm²). Clearly, the growth of VGs and GFs can substantially enhance the density of exposed active sites accounting for a larger ECSA than that of pristine carbonized wood, because the surface roughening, microstructure reconstruction and amorphization during the electrochemical activation enable the exposure of more active sites.

Figure 15. CV curves of a) 3D-LC, b) 3D-RC, c) 3D-LFGC and d) 3D-RFGC electrodes at various scan rates ranging from 5 to 100 mV s⁻¹ in 2 M ZnSO₄. (e) The linear fitting of Δj (difference between anodic and cathodic current densities) versus scan rate for ECSA estimations of the various electrodes.

As shown in **Figure 16**, the 3D-LC and 3D-RC show an ECSA of 26 and 62.5 cm², respectively. The ECSA of the 3D-RFGC matrix increases to 392.5 cm², higher than that of the 3D-RFGC matrix (312.75 cm²). The comparable ECSA for 3D-RFGC can be ascribed to the introduction of GFs and VGs with enlarged specific surface area and abundant active sites.

Figure 16. Quantitative comparison of double-layer capacitance and calculated ECSA results.

Comment #4: *How did the authors determine the voltage hysteresis? At which capacity did the authors record the voltage? Is the voltage hysteresis in this work the sum or the average of the anodic and cathodic processes? Please add these details to the supporting information. I also recommend adding voltage hysteresis in Table S1 and S2.*

Response: We have added the details of voltage hysteresis in the revised supporting information and the analysis and discussions are as follows:

For metal anodes (for instance, Li metal, Zn metal), the electrochemical behavior of Zn plating/stripping and the cycling stability were examined by comparing the galvanostatic discharge/charge voltage profiles in symmetric Zn/Zn cells. The voltage polarization, also called voltage hysteresis, is the difference between the voltages of Zn stripping and plating and is mainly determined by the current density, interfacial properties and charge transfer resistance (*Adv. Funct. Mater.* 2020, 30, 2000786; *Nat Commun* 2020, 11, 93; *Adv. Sci.* 2019, 6, 1901776).

We take for example the plating/stripping curve of the 3D-RFGC@Zn anode in symmetric cells, ΔU_1 is considered to be voltage hysteresis according to previous reports. As shown in **Figure 17**, the hysteresis in this manuscript is the voltage difference of the plating and stripping processes. The voltage hysteresis of the entire cycling also indicates large voltage fluctuations in two areas, and gradually resumes the

normal deposition and stripping process (**Figure 18**). This phenomenon still exists under higher current densities and capacities. However, we can still confirm that it is not a short circuit through the enlarged deposition/stripping curves. Also, we added voltage hysteresis information in **Table S1** according to your advice.

Figure 17. Schematic illustration of polarization and voltage hysteresis in symmetrical cells.

Figure 18. The hysteresis of the entire cycling to show the polarization in symmetric cells.

Table S1 Comparison of electrochemical performances of our 3D-RFGC@Zn anode and reported materials for ZMAs. (C_d : current density; C: capacity; L:lifespan, V_h : voltage hysteresis)

Samples	C_d (mA cm ⁻²)	C (mAh cm ⁻²)	L (h)	V_h (mV)	Reference
CuZIF-L@TM	6	7	650	80	14
TZNC	1	1	450	12	15
3D Ni-Zn	5	2	300	/	27
MXene/Graphene	10	1	1050	64	16

Aerogel					
CnC HS	4	1	116	40	17
3D Zn	10	1	190	45	18
3D Ti-TiO ₂	10	0.5	500	/	19
3DGT@Zn	2	1	1100	/	20
3D SnPCF	10	5	500	47	21
Zn/CNT	5	2.5	110	68	23
Cu-Ps/EG	10	10	3000	71.4	24
AgNWA@Zn//C-MOF	40	10	90	/	25
3D-Topo-Zn	2	2	1160	/	26
3D-ZGC	20	1	150	65	28
3DP-ZA	4	2	180	/	29
Zn@PCH	10	1	110		22
SS-V ₂ O ₃ @C/Zn	20	20	5000	/	30
3D-Zn	5	5	160	/	31
Ti ₃ C ₂ Tx	10	1	180	/	32
MXene/ZnS					33
N-VG@CC	1	1	70		33
FPCH-ZI/Zn	10	5	650	43.5	34
3D-RFGC@Zn	1	1	7300	15	This work
3D-RFGC@Zn	20	20	2000	32	This work
3D-RFGC@Zn	40	40	2200	26	This work
3D-RFGC@Zn	80	80	2600	31	This work

Comment #5: *In Figures 3b and 3c, why are there sharp turning points in the curves? If the sampling intervals are too wide, the calculated coulombic efficiencies might not be accurate.*

Response: We double checked the CE data and also conducted the CE tests under different current densities from 1 to 120 mA h cm⁻² with a fixed capacity of 0.5 mA h cm⁻². However, the issue of the sharp turning points in the CE curves you mentioned still exists at low deposition capacity of Zn. As an example, the deposition time decreases with the increase of deposition current density (1~120 mA cm⁻²), especially when the deposition current exceeds 40 mA cm⁻², it takes only 0.025 h (1.5 seconds) to deposit. Therefore, because of the limited precision of the battery test equipment, only

little data can be recorded in this situation, resulting in sharp turning points in the CE curves. As can be seen from the curve below, to get a smoother CE curve, we increase the capacity of Zn deposited (5 mAh), thus prolonging the deposition time.

Figure 19. The CE curve of 3D-LFGC at 40 mA cm^{-2} with a capacity of 5 mAh.

Comment #6: *In Figure 3j, the authors did not compare the data with 3D-LFGC.*

Response: We have added the over-potential data of 3D-LFGC in CE test in Figure 3j, and data of over-potential corresponding to 3D-LFGC and 3D-RFGC are summarized into Figure 3k.

Comment #7: *In Figure 3l, are the labels for 3D-LFGC and 3D-RFGC reversed?*

Response: We are sorry that 3D-LFGC and 3D-RFGC were marked reversely in Figure 3l and the error has been modified in the revised manuscript.

Comment #8: *The reviewer suggests that the authors double-check the data of this work in Figure 3m.*

Response: We have double checked the data provided in Figure 3m, and further modified the drawing and data in the revised manuscript, shown in Figure 3l and Table S2.

Comment #9: *In the supporting information of 3D-FGC@Zn synthesis, the authors state that “High areal capacities, such as 5 or 10 mAh cm⁻², were also conducted to increase the deposition rate”. Did the authors mean high areal current densities such as 5 or 10 mA cm⁻²?*

Response: We did cause a writing error in the synthesis of the 3D-FGC@Zn composite anodes. Details are as follows:

(1) The 3D-LFGC@Zn and 3D-RFGC@Zn anodes were prepared through the electrochemical deposition method. Zinc foil (150 μm) was used as the zinc source. Zn metal was deposited on Cu foil, 3D-LFGC and 3D-RFGC matrices via coin cells. The current density of depositing Zn was set to be 1 mA cm^{-2} , and the capacity was set to from 1 to 80 mAh cm^{-2} (including 1, 2, 5, 10, 20, 30, 40, 80 mAh cm^{-2}).

(2) If only low current density (1 mA cm^{-2}) is used, it will take a long time to obtain a zinc anode with high capacity. Higher current densities, such as 5 mA cm^{-2} and 10 mA cm^{-2} , can be used to accelerate the deposition rate of Zn metals. For example, a 40 mAh cm^{-2} zinc anode can be obtained with a current density of 1 mA cm^{-2} for 40 h, while with a current density of 10 mA cm^{-2} for only 4 h.

Figure 20. The curves of depositing Zn metal on the 3D-RFGC matrices.

Comment #10: *The data of bare Zn was not presented in Figure 4f.*

Response: We have checked all the electrochemical measurement data, including CE, symmetrical cells, and full cells again. The bare Zn suffers from short-circuit at the beginning of plating when the current densities are beyond 10 mA cm^{-2} . As shown in **Figure 21**, we show the plating and stripping curves of bare Zn under different current densities, short-circuit occurs quickly with the increase of current densities and capacities of plating and stripping. According to your comments, we have added the data in *Figure 4f* as the benchmark. Also, we have included the data of symmetrical cells for 3D Zn foam as the benchmark in *Figures 4a, 4i and 4j*.

Figure 21. Voltage profiles of Zn metal plating/stripping at different current densities and different areal capacity. a) $10 \text{ mA cm}^{-2}/10 \text{ mAh cm}^{-2}$, b) $20 \text{ mA cm}^{-2}/20 \text{ mAh cm}^{-2}$, c) $40 \text{ mA cm}^{-2}/40 \text{ mAh cm}^{-2}$ and d) $80 \text{ mA cm}^{-2}/80 \text{ mAh cm}^{-2}$.

Comment #11: *The charge/discharge curves of 3D-RFGC@Zn symmetrical cells fluctuate significantly at all rates, especially in Figure S14. Also, the overpotential at 1 mA cm^{-2} is much higher than those at other rates, the overpotential at 10 mA cm^{-2} is much lower than that at 5 mA cm^{-2} , and the overpotential at 60 mA cm^{-2} is much lower than that at 30 mA cm^{-2} , which do not make sense. Please explain these findings.*

Response: Based on your comments, we have further double-checked the Zn/Zn symmetrical cells data in this manuscript, the voltage curves of 3D-RFGC@Zn and 3D-LFGC@Zn symmetrical cells fluctuate significantly at all rates, especially under the higher current densities and capacities.

(1) We have added the voltage curves at different durations to show the polarization in symmetric cells (**Figure 22**). As you can see, although there is a voltage fluctuation, the enlarged curves of the depositing/stripping curves are still normal in the whole process of testing (**Figures 22b-22e**), which also exists in other reports¹³⁻¹⁵. Moreover, we can still confirm that it is not a short circuit through the enlarged deposition/stripping curves.

Figure 22. The enlarged voltage curves (a-e) at different stages and (f) hysteresis of the entire cycling to show the polarization in symmetric cells.

(2) We are so sorry to make some mistakes in labeling test conditions (current density and capacity) to data in Figure S14 after re-checking all symmetrical cells data. Therefore, we re-conducted the symmetrical cells at different current densities and capacities, including 820 h for $2 \text{ mA cm}^{-2}/2 \text{ mAh cm}^{-2}$, 400 h for $5 \text{ mA cm}^{-2}/5 \text{ mAh cm}^{-2}$, 400 h for $8 \text{ mA cm}^{-2}/8 \text{ mAh cm}^{-2}$, 200 h for $20 \text{ mA cm}^{-2}/20 \text{ mAh cm}^{-2}$, respectively.

Figure S23. Galvanostatic cycling of 3D-RFGC@Zn symmetrical cells at 5 mA cm^{-2} and 5 mAh cm^{-2} , 8 mA cm^{-2} and 8 mAh cm^{-2} , 20 mA cm^{-2} and 20 mAh cm^{-2} .

Comment #12: *The reviewer suggests a table summarizing the fitted EIS data.*

Response: According to your suggestion, we have added three tables to summarize the

EIS data, including *Figure 5k*, *Figure 7e* and *Figure S26* in the revised manuscript.

Details are as follows:

Table S3. The values of R_{ct} of in-situ EIS curves of 3D-RFGC@Zn/3D-RFGC@Zn and bare Zn/Zn symmetric cells during continuous Zn plating process. (C_n -cycle number (n); R_{ct} -charge transfer resistance)

C_n (n)	R_{ct} (Ω)	C_n (n)	R_{ct} (Ω)
3D-RFGC@Zn/3D-RFGC@Zn		Zn/Zn	
1	182.54	1	622.32
2	192.64	2	658.97
4	190.46	4	720.89
6	214.47	6	782.64
8	190.50	8	907.09
10	204.93	10	1008.6
15	200.53	15	1192.8
20	203.95	20	1524.3
30	214.47	25	3914.9
40	218.04	30	4194.3
50	228.32	35	4862.3
55	231.86	40	5333.5
70	247.92	45	6165.3
/	/	50	8712.5
/	/	55	10210.0

Table S4. The values of R_{ct} in-situ EIS spectra of $V_2O_5@3D-LC//3D-RFGC@Zn$ full cells electrode before cycling and cycling at different charge/discharge stages.

C_n (n)	R_{ct} (Ω)	C_n (n)	R_{ct} (Ω)
1	200.61	16	27.94
2	196.98	17	26.07
3	190.25	18	24.61
4	172.95	19	23.29
5	174.91	20	22.18
6	161.88	21	21.28
7	133.95	22	20.48
8	94.46	23	18.95
9	70.35	24	18.09
10	56.16	25	17.61
11	46.92	26	17.37
12	40.76	27	16.91
13	36.45	/	/
14	33.05	/	/
15	30.10	/	/

Comment #13: *How exactly does 3D hierarchical graphene matrix protect Zn anode from corrosion, given that the 3D hierarchical graphene matrix would increase the accessible area of Zn? By reducing dendrites?*

Response: We explain the protection mechanism of 3D hierarchical graphene matrices on zinc metal anode as follows:

Recently, Zn host construction has been considered to be effective in suppressing the prevalence of dendritic Zn and the formation of dead Zn for the long-term cycling of ZMBs under harsh operating conditions. A Zn host with a large SSA can induce a uniform current density owing to the uniform distribution of the local current density and Zn^{2+} ions concentration distributions. Additionally, because the Zn deposits are confined to large active spaces, the formation of dead Zn can be alleviated, and restricts the volume fluctuation of the Zn anode and degradation of the CE for repetitive Zn

deposition/dissolution. Specifically, our manuscript contains four aspects to explain the protection strategy for zinc metal:

i) The pristine 3D multichannel carbon matrices provide continuous pore structure with sufficient spaces for zinc metal deposition. Meanwhile, the vertically oriented structure can shorten the ion diffusion path and improve the reaction kinetics (**Figures 24a-24d**).

ii) N-doped graphene nanofibers clusters (GFs) and vertical graphene arrays (VGs) modified multichannel carbon (3D-FGC) are for the first time achieved by a one-step thermal CVD method, providing enlarged specific surface area (**Figure 25a**) and abundant zincophilic sites (**Figure 25b**), thereby reducing the local current density, homogenizing electric field distribution and ion concentration gradient.

iii) After fabricating the layer of GFs and VGs, the Zn^{2+} flux in this architecture with abundant zincophilic sites is effectively homogenized with bottom-up deposition. The whole electrode surface gets smooth with uniform and compact Zn deposition and the whole empty space is gradually filled as the deposition time increases (**Figure 26**).

iiii) The corrosion potential of 3D-RFGC@Zn (-0.958 V), 3D-LFGC@Zn (-0.983 V) was higher than the bare Zn electrode (-1.014 V) (**Figure 27a**). Moreover, after depositing/stripping certain cycles in $ZnSO_4$ electrolyte, massive by-products, $Zn_4SO_4(OH)_6 \cdot xH_2O$ (Advanced Functional Materials 32, (2022); Advanced Energy Materials 11, (2021); ACS Energy Letters 4, 2776-2781 (2019)) are generated due to severe corrosion for bare Zn electrode, whereas there are nearly no obvious by-products on 3D-RFGC@Zn electrode according to the results of ex-situ XRD patterns (**Figure 27b**). This result indicates that the 3D hierarchical graphene matrices can protect Zn anode from corrosion. From **Figures 27c-27d**, a great mass of macroscopic passive aggregates is continuously forming and detaching from the Zn surface. In sharp contrast, the 3D-RFGC surface maintains a smooth morphology with no clutter observed. The reason is that the 3D graphene host is helpful to the uniform deposition of Zn metal, which makes it possible to obtain a dense and stable electrolyte/Zn interface, thus slowing down the Zn corrosion and other side reactions.

Figure 24. SEM images of pristine a, b) 3D-LC and c, d) 3D-RC.

Figure 25. a) The SSA of different carbon matrices and Zn anodes; b) Binding energy of the pristine graphene and N-doped graphene.

Figure 26 The distributions of deposited Zn on a) 3D-LFGC, b) 3D-RFGC, and corresponding SEM images of c) 3D-LFGC and d) 3D-RFGC, respectively.

Figure 27. (a) Linear polarization curves showing the corrosion on bare Zn and 3D-Zn. (b) XRD patterns of different Zn anodes after 10 cycles and 50 cycles. Three-electrode system of (c) bare Zn plates and (d) 3D-RFDC host in transparent tanks representing the side reactions visually during continuous Zn plating/stripping at a current density of 5 mA cm².

Comment #14: *It seems like the active surface of 3D-RFGC is just the top quasi-2D layer, which means the electrolyte and ions cannot penetrate the top layers to access the bottom.*

Response: We have explained the penetration of electrolyte in the 3D-RFGC electrode as follows:

(1) Undoubtedly, in our models and 3D matrices, the diffusion and deposition of zinc ions are three-dimensional, and the obtained composite zinc anodes are also three-dimensional, which can be proved from the material structure. As shown in **Figure 28a**, for 3D-LC, the vertically oriented pore structure is conducive to the direct deposition of zinc metal, and it is obvious that 3D zinc anode will be obtained. For the pristine 3D-RC (**Figure 28b**), there is still a pore structure between the carbon channels to ensure the penetration of the electrolyte along the vertical direction. Building a GFs and VGs layer with a thickness of not less than 20 microns on the electrode is extremely beneficial for regulating the current distribution and zinc ion concentration, so as to ensure the uniform deposition of zinc metal on the skeletons (**Figure 28d**).

(2) 3D current collectors minimize the tortuosity of the ion and electron transport pathways, especially in highly loaded electrodes. According to the Fick's law, the ion diffusion time $\tau = L^2/2D$, where D is the diffusion coefficient, and L is the diffusion length. When using 3D-RFGC electrode, the diffusion time is proportional to the diffusion distance. Therefore, 3D-RFGC has larger diffusion resistance and longer diffusion time than the 3D-LFGC does.

(3) According to experimental and simulation results, with the increase of diffusion distance, the Zn plating/stripping kinetics slows down at higher deposition capacity, resulting in the reduction of plating and stripping efficiency. In the 3D-RC and 3D-RFGC models, there is a relatively low diffusion distance, thus ensuring a high coulomb efficiency of zinc plating and stripping (**Figure 29**). Simulation results of the current density distribution indicate that the areas with high reactivity are concentrated in a certain thickness, and the reaction of the whole electrode is actually limited, which can also be further proved by the electrochemically active surface area, as introduced in the response of comment 3.

Figure 28. SEM images of a) 3D-LC, b) 3D-RC, c) 3D-LFGC and d) 3D-RFGC matrices.

Figure 29. Simulation results of the current density distribution of a) 3D-RFGC, b) 3D-LFGC, c) 3D-RC and 3D-LC matrices.

Comment #15: *The reviewer doubts the correctness of the model for 3D-LFGC. From the SEM images (Figure S5), the channel openings are much larger than the top surface.*

Response: It is true that the area of channel openings is larger than the top surface for 3D-LFGC, but electrochemical reaction rate of channel openings is smaller than that of top surface. We have neglected the reaction sites of channel opening. Meanwhile, to enhance the accuracy of this model, we have added the electrochemical reaction of channel opening. According to the COMSOL simulations, we established models to prove the action of a multi-function layer of GFs and VGs in regulating current density and Zn-ion flux distributions, as depicted in **Figure 30**. In these models, we added the electrochemical reaction of channel openings and graphene layers.

Figure 30. The geometrical structures and boundaries condition for (a) 3D-LC; (b) 3D-LFGC; (c) 3D-RC and (d) 3D-RFGC.

Comment #16: *Following question #15, if the worse performance of 3D-LFGC came from the more accessible top surface and less accessible bottom of channels, would it be better to choose a woodblock precursor with larger channel sizes? What kind of natural wood block did the authors choose and why?*

Response: Combining the experimental results and simulation data, we explain the reasons as follows:

(1) Compared with 3D-RFGC, 3D-LFGC has a larger thickness of electrode (500 μm) than that of 300 μm , which is unfavorable to the diffusion and transport of zinc

ions. Although the wettability of electrolyte is enhanced by the introduction of GFs and VGs layer, the mass transfer resistance of zinc ions also increases with the increasing thickness. From SEM images of 3D-LC@Zn (**Figure 31a-31b**) and 3D-LFGC@Zn (**Figure 31d-31f**), we can find that Zn metal deposited in the channels of 3D-LFGC when introducing GFs and VGs. In contrast, zinc tends to deposit on the upper surfaces of 3D-LC, blocking the channels and thus reducing channel utilization and leading to dendrite formation. According to the COMSOL simulation results, after fabricating the layer of GFs and VGs, the Zn^{2+} flux in this architecture with abundant zincophilic sites is effectively homogenized with bottom-up deposition. The top of 3D-LC showed lower zinc ion concentration distribution than other areas, thus leading to uneven Zn deposition and dendrites growth with “tip effect” (**Figure 31c**). The results of 3D-RFGC indicate uniform Zn ion distribution and fast reaction rate in the channels because of the graphene layers (**Figure 31f**). However, the deterioration of coulombic efficiency and the ex-situ SEM images show that the zinc existing in the multi-channel is gradually difficult to strip during the repeated Zn deposition and stripping process, and gradually accumulated and developed into dendrites (**Figure 32**).

Figure 31. SEM images of a, b) 3D-LC@Zn and d, e) 3D-LFGC@Zn anodes. The corresponding COMSOL simulation results of c) 3D-LC@Zn and f) 3D-LFGC@Zn.

Figure 32. SEM images of a) 3D-LFGC@Zn anodes.

(2) We believe that the key to the issue is not to choose a woodblock precursor with larger channel sizes, but to reduce the thickness of 3D host and regulate the surface characteristics of the 3D matrices, such as surface modification. It is beneficial to improve the wetting property of electrolyte, shorten the Zn^{2+} ions transport pathways and accelerate the diffusion process to improve the utilization of 3D current collector, enhance kinetics and maintain good stability. According to our simulation results, the reaction rate on the surface of the 3D matrices is higher than that of the channel openings. Therefore, only increasing the pore size of 3D matrices is unfavorable to the deposition and stripping of zinc metal. In previous literature, there are many kinds of wood used as precursor of porous materials, (for instance, basswood, cherry, pine, walnut, maple, hickory, paduak, tigerwood and ipe). We also investigate four samples (camphor-wood, beech-wood, basswood, pinewood) to compare the morphology of different woods after carbonization, as shown in **Figure 33**. All samples possess large porous structures and irregular shapes. Finally, we selected basswood in this manuscript based on many previous reports in electrochemical energy storage and conversion.

Figure 33. SEM images of a) camphor-wood, b) beech-wood, c) basswood, and d) pinewood after carbonization.

Comment #17: *Following question #16, the authors claim that a high channel of 800μm is not conducive to the complete stripping of zinc metal. However, there are many successful works reported for thick 3D Zn anodes (Adv. Energy Mater. 2021, 11, 2003927; Science 2017, 356(6336), 415; etc.).*

Response: The literature you mentioned has indeed achieved excellent performance, but in many works, what we call “incomplete stripping” also exists. Constructing 3D porous zinc anodes has proven to be an effective way to significantly improve their electrochemical performance. However, there are still many factors that affect the performance of AZIBs, such as material composition, pore structure and size, specific surface area, preparation process, etc.

In this manuscript, two 3D carbon matrix models are proposed to study the deposition and stripping behavior of zinc. The thickness of 3D-LFGC in the manuscript is 500 μm, and the thickness of 3D-RFGC is 300 μm. The performance of 3D-RFGC is better than that of 3D-LFGC, according to CE, symmetrical cells tests and COMSOL simulations. The performance of 3D-LFGC decays after repeated cycling. We can find that the residual zinc metal exists in the carbon channel by ex-situ SEM, so we described it as “incomplete stripping”. This result is mainly because that the electrolyte was consumed and the ion diffusion resistance increases in subsequent processes, which

leads to unstable deposition and stripping efficiency, especially at high current density.

Comment #18: *What are the differences between the models of carbonized wood with and without vertical graphene arrays and graphene nanofibers? The models given in this work are indistinguishable from each other.*

Response: To enhance the accuracy of this model, we have added additional improvements and explanations in *Supporting Information* to better understand our model.

A 3D transient model was developed to numerically resolve coupled multi-physical including ion transport and chemical reactions. Specifically, four models (including 3D-LC, 3D-LFGC, 3D-RC, 3D-RFGC) are modeled, as shown in **Figure 30**. In all models, the overall size is set to $20 \times 20 \times 10 \mu\text{m}$ (length \times width \times height), the diameter of the pristine carbon channel is set to $6 \mu\text{m}$, and the wall thickness is set to $1 \mu\text{m}$, the thickness of the graphene functional layer is also set to $1 \mu\text{m}$. To enhance the accuracy of this model, we have added the electrochemical reaction of channel openings. Simulated Zn^{2+} ion concentrations and current distributions are conducted to analyze the deposition and stripping behavior of Zn. The simulation parameters and operation details are as follows:

1) Governing equations

The conservation for the ion in the electrolyte domain concludes times derivative term, electromigration and diffusive transport, and ion consumption by chemical reactions.

$$\frac{\partial c_i}{\partial t} + \nabla \cdot (-D_i \nabla c_i) + \nabla \cdot (-z_i u_i F c_i \nabla V) = R_i \quad (1)$$

where $R_i = \frac{-i_{\text{loc}}}{2F}$ is the electrochemical reaction source term, i_{loc} is the local volume current density, $D_i = \frac{1}{\varepsilon} D_0$ is the effective diffusion coefficient ($D_0 = 1 \times 10^{-6} \text{ cm}^2/\text{s}$ is the intrinsic diffusion coefficient), $\varepsilon = 0.5$ is the porosity for porous electrode ($\varepsilon = 1$ for the electrolyte domain), u_i is the mobility (defined by the Nernst-Einstein equation), c_i is the ion concentration, $z_i = 2$ is the charge number, F is the Faraday constant, and V is the electric potential.

The Butler-Volmer equation was used to describe the Zn plating process and the local

current density was as a function of potential and ion (Zn^+) concentration:

$$i = i_0 \left(\exp\left(\frac{1.5F\eta}{RT}\right) - \frac{c_i}{c_i^0} \exp\left(-\frac{0.5F\eta}{RT}\right) \right) \quad (2)$$

where i_0 is exchange current density, η is the overpotential, c_i^0 is the initial Zn^+ concentration. Note that the active specific surface area of the electrode is 10^{-9} m^{-1} .

2) Boundary conditions

Figure S32 shows the boundary condition of two models. The initial concentration of Zn^{2+} is 10 mol/m^3 . The current density i and Zn^{2+} concentration at the top of bulk solution is 100 mA/cm^2 and 10 mol/m^3 , respectively. Note that all electrode surfaces and porous electrodes domains are the electrochemical reaction sites. The potential of the electrode surface is 0 V , and the rest boundaries are insolution. Besides, the rest boundaries for the ion transport is no flux.

Comment #19: *What are the Zn anode's loadings for the full device tests? The authors used gravimetric current densities and capacities. Were they calculated based on the mass of active cathode materials? Please also specify the mass loading of active cathode materials. What are the corresponding areal current densities, areal capacities, and depths of discharge of the Zn anode?*

Response: We have added the information you mentioned in the revised supporting information. The reason can be explained it in detail here.

1) In full cells, the 3D-RFGC@Zn anodes with a capacity of 40 mAh cm^{-2} were obtained by electrochemical deposition method in half-cells. Zn foil with a thickness of $150 \text{ }\mu\text{m}$ and diameter of 14 mm was used as zinc source. The current density of depositing Zn was set into 1 mA cm^{-2} , to ensure uniform deposition of zinc metal.

2) The current capacities and capacities of $\text{V}_2\text{O}_5@3\text{D-LC}/3\text{D-RFGC}@Zn$, $\text{MnO}_2@3\text{D-LC}/3\text{D-RFGC}@Zn$ and $\text{AC}@3\text{D-LC}/3\text{D-RFGC}$ were calculated based on the mass of active cathode materials. For $\text{V}_2\text{O}_5@3\text{D-LC}/3\text{D-RFGC}@Zn$ cathodes, we prepared the different mass loading of $\text{V}_2\text{O}_5@3\text{D-LC}$ cathodes, including 36.2 , 22.6 , 12.5 and 1.6 mg cm^{-2} . For $\text{MnO}_2@3\text{D-LC}$ cathodes, we prepared the different mass loading of $\text{MnO}_2@3\text{D-LC}$ cathodes, including 35.0 , 22.6 , 12.5 and 1.0 mg cm^{-2} . For $\text{AC}@3\text{D-LC}$ cathodes, we prepared the different mass loading of $\text{AC}@3\text{D-LC}$ cathodes.

The corresponding areal current densities, areal capacities, and depths of discharge of the Zn anode are also marked and illustrated in *Supporting Information*.

Comment #20: For 3D V_2O_5 cathodes, it seems like V_2O_5 was loaded on 3D-LFGC. Please specify this in the main text and consider changing the name to $V_2O_5@3D-LFGC$. $V_2O_5@3DG$ in the main text is ambiguous. Why didn't the authors load V_2O_5 on 3D-RFGC?

Response: Based on your suggestion, we have modified the nomenclature of the $V_2O_5@3D-LC$, $MnO_2@3D-LC$ and $AC@3D-LC$ cathodes in the revised manuscript. The detailed interpretation is described as follows.

(1) During preparing V_2O_5 cathodes, inspired by the unique vertically-aligned multichannel structure of the carbonized-wood materials, we fabricated the $V_2O_5@3D-LC$ cathode with ultrahigh mass loading and low tortuosity by simply infiltrating the V_2O_5 materials into the channels of the 3D-LC (**Figure 34**). The 3D $V_2O_5@3D-LC$ electrode design enables more electroactive materials, thus obtaining higher mass energy densities.

Figure 34. The device of vacuum filtration and SEM images of $V_2O_5@3D-LFGC$ cathodes with mass loading of a, a1) 4 mg cm^{-2} , b, b1) 12.5 mg cm^{-2} , c, c1) 22.6 mg cm^{-2} , and d, d1) 35.0 mg cm^{-2} .

(2) We also load V_2O_5 cathode on 3D-RC to match with 3D-RC@Zn. However, limited mass loading and rapid capacity decaying indicate that 3D-RC is not a good choice for a cathode current collector. The cracking and shedding of the cathode material are more serious at high loading of V_2O_5 (**Figure 35**). In 3D-LC, the active materials are restricted in a multichannel carbon matrix, which ensures both high mass loading and structural stability of the electrode during the cycles.

Figure 35. The SEM images of $V_2O_5@3D-RC$ with a, b) high mass loading and c, d) low mass loading of V_2O_5 materials.

Comment #21: *Were MnO_2 and AC cathodes also 3D? The authors didn't mention this in either the main text or supporting information.*

Response: The data of $3D-RC@Zn/MnO_2$ full cells and $3D-RC@Zn/AC$ capacitor provided in this manuscript are not 3D cathodes. Cathodes are prepared by the traditional method of casting and drying. Specifically, the MnO_2 cathode was prepared by mixing cathode powder, Ketjen black (KB), and polytetrafluorethylene (PTFE) in a mass ratio of 75:15:10 by isopropyl alcohol, and then casting such slurry on titanium foil. After drying at 80 °C overnight in a vacuum oven, the electrodes with 1.5~2.0 mg cm^{-2} MnO_2 were achieved.

We also conducted the experiments of $3D-RFGC@Zn/MnO_2@3D-LC$ full cells and $3D-RFGC@Zn/AC@3D-LC$ capacitor to improve the logic and integrity of the whole research. As shown in **Figure 36** and **Figure 37**, activated carbon (AC) and MnO_2 inks were infiltrated into the 3D-LC matrices. The carbon channels were filled with active cathode materials as time increased, thus obtaining the $MnO_2@3D-LC$ and $AC@3D-LC$ with different mass loadings. High discharge areal capacities can be achieved through the cycle curves and charge-discharge curves (**Figures 38-39**).

Figure 36. SEM images of AC@3D- LC at a, b) low mass loading and c, d) high mass loading.

Figure 37. SEM images of MnO₂@3D-LC at a, b) low mass loading and c, d) high mass loading.

Figure 38. Cycle performance of 3D-RFGC@Zn/AC@3D- LC capacitor with different mass loading of AC.

Figure 39. Charge-discharge curves of 3D-RFGC@Zn/V₂O₅@3D- LC full cells with different mass loadings of V₂O₅.

We want to thank the reviewers for the valuable comments and suggestions, which are helpful for improving the quality of this manuscript.

REVIEWER COMMENTS

Reviewer #1 (Remarks to the Author):

The authors have spent good effort to improve the manuscript and addressed most concerns raised by the reviewers. However, the revised manuscript continues to emphasize high areal capacity one-sidedly while omitting the weakness in the volumetric capacity. The reviewer notes that the two indices should not be seen as two sides of a coin that are somehow difficult to achieve simultaneously; quite the opposite, thick electrodes are often pursued as a means towards high volumetric energy. While the authors are free to decide how they want to address this area, the review thinks a comparison of the space utilization in the "ZMAs" reports that the authors are already comparing.

Editorial Note: Here Fig 12 and 14 in Reviewer #2's report refers to the corresponding figures in the authors' point-by-point response to the Reviewer's previous report.

Reviewer #2:

Remarks to the Author:

The authors answered to my main concerns. Nonetheless, still some points remain to be improved:

1) This is fine

2) The color doesn't match between the caption and data of Figure 12a. In addition, the main concern is that the scale of y-axis is different in Figure 12b-e. Please match the y-axis scale to be same (maximum voltage to be 0.1 and minimum to be -0.1). Furthermore, I still doubt that the performance of your data does really exceed the other recently published papers in Nature Communications.

3) This is fine

4) Same as review #2, I recommend you to match the scale of y-axis of Figure 14a-d, and it's better to use “,” than “~” for areal capacity. I wonder in figure 14a, the step time seems to be changed before and after 10 hours. I think you have to support about the difference. Also, as I understand your answer, the cell short-circuit occurred only in figure 14d, you have to mention about your criteria of cell short-circuit. At last, you need to explain more about the voltage curve fluctuation during the cycle, because it occurs at every current density in figure 14a-d.

5) This is fine

6) This is fine

7) This is fine

Due to the comments above, I suggest still a major revision, in order to fulfill the aforementioned points of 2 and 4.

Reviewer #3 (Remarks to the Author):

The reviewer appreciates the authors' efforts and is satisfied with most responses and revisions. I do have a few more minor comments for the authors to consider. After adequately addressing these comments, the manuscript should be suitable for publication.

1. Fig. 3I needs to be simplified and clarified. For example, it took me a while to find the correct y-axis to read the values of the columns or data points.

2. In Fig. 4I, the authors use the size of circles to represent the values of areal capacities, which needs to be clarified. I suggest putting a scale bar so that the readers can measure the diameters of the circles, or change to other clearer representations. I also recommend putting all the data of Fig. 4I in a table in the supporting information.

3. Looking at the schematic illustration of Fig. 7a, the readers might think the authors use the 3D-LFGC@Zn anode, while in fact, the electrode is 3D-RFGC@Zn. Please correct this misleading figure.

Dear reviewers:

Thank you very much for your time and work on our manuscript. Your positive and constructive comments are important for the improvement of our manuscript. According to your further comments, we improved the manuscript, which was marked with red color in the revised manuscript. The modifications in our manuscript are described below.

Responses to the comments by Reviewer #1

Reviewer #1: The authors have spent good effort to improve the manuscript and addressed most concerns raised by the reviewers. However, the revised manuscript continues to emphasize high areal capacity one-sidedly while omitting the weakness in the volumetric capacity. The reviewer notes that the two indices should not be seen as two sides of a coin that are somehow difficult to achieve simultaneously; quite the opposite, thick electrodes are often pursued as a means towards high volumetric energy. While the authors are free to decide how they want to address this area, the review thinks a comparison of the space utilization in the "ZMAs" reports that the authors are already comparing.

Response: Thank you for your further comments. We have to admit that in our study there is a lack of space utilization to obtain a low volumetric capacity. However, in our manuscript, the high operating current density and the high deposition/stripping capacity of zinc are the advantages of the study, but most of zinc deposition/stripping reaction occurs on the upper surface of 3D-RFGC, as shown in **Figure 1**.

From our COMSOL simulations, we can conclude that for a 3D host, the current density applied during the reaction is fixed (40 mAcm^{-2}), large specific surface area and 3D space can reduce the local current density, which is one of the reasons why 3D-RGFC can work continuously at high current density and high surface capacity. To a certain extent, the current electrode thickness ($300 \text{ }\mu\text{m}$) for 3D-RGFC is indeed disadvantageous to improve the volumetric capacity of the cells, while it has a nonnegligible benefit for the cells to work under the high current densities and capacities.

Figure 1. Schematic illustration of deposited Zn distribution on 3D-RFGC host.

To this end, we also provide two solutions, one is to reduce the thickness of the electrode through simple sandpaper grinding or other advanced electrode thinning technology; the other is to make more channels on the 3D current collector artificially by precision machining technology, such as laser drilling, which is convenient for zinc deposition and stripping in the 3D space to improve the utilization of the 3D space. We believe that both of these strategies are effective in increasing the volumetric capacity of the whole electrode.

Based on the present situation, we have also made further revisions to the manuscript regarding your concerns about high surface capacity and space utilization. First, we do not emphasize the issue of high-areal-capacity in the manuscript, but instead, modify the statement that *surface modifications of vertical graphene arrays (VGs) and graphene nanofiber clusters (GFs) regulates surface high-capacity zinc deposition and stripping*". Second, we have tried to compare 3D-RFGC with other reported 3D hosts for measuring the space utilization of zinc anodes, but this parameter is more difficult to measure accurately. At present, there are few references on the work of zinc anodes. We have made an approximate trend graph in terms of the utilization of 3D space (**Figure 2**) according to the 3D hosts for lithium anodes reports (*Small* 2022, 18, 2106718; *Angew. Chem. Int. Ed.* 2019, 58, 3092-3096; *Adv. Funct. Mater.* 2022, 32, 2112151; *Nano Energy* 2020, 69, 104471). Third, high zinc utilization will be highlighted as an innovation in the revised manuscript. With the introduction of VGs/GFs layer, these Zn anodes have exhibited significantly enhanced cycling stability

and efficiency under high current densities with 100% DOD (Depth of Discharge). Meanwhile, as you requested, we have included the comparison of the DOD and space utilization of Zn anodes in the revised **Table 1**.

Figure 2. The comparison of the space utilization of 3D hosts in the "ZMAs" reports.

Table 1. Comparison of electrochemical performances of our 3D-RFGC@Zn anode and reported 3D materials for ZMAs. (C_d : current density; C: capacity; L:lifespan, V_h : voltage hysteresis; DOD: depth-of-discharge; U_s : space utilization)

Samples	C_d (mA cm^{-2})	C (mAh cm^{-2})	L (h)	V_h (mV)	DOD (%)	U_s (%)	Reference
CuZIF-L@TM	6	7	650	80	50	~10	1
TZNC	1	1	450	12	50	~60	2
3D Ni-Zn	5	2	300	/	/	~5	3
MXene/Graphene Aerogel	10	1	1050	64	60	~40	4
CnC HS	4	1	116	40	/	/	5
3D Zn	10	1	190	45	/	~30	6
3D Ti-TiO ₂	10	0.5	500	/	/	~70	7
3DGT@Zn	2	1	1100	/	47.1	~30	8
3D SnPCF	10	5	500	47	/	~60	9
Zn/CNT	5	2.5	110	68	28	~60	10
Cu-Ps/EG	10	10	3000	71.4	/	~70	11
AgNWA@Zn//C-MOF	40	10	90	/	/	~50	12
3D-Topo-Zn	2	2	1160	/	11.4	/	13
3D-ZGC	20	1	150	65	/	/	14
3DP-ZA	4	2	180	/	/	~40	15
Zn@PCH	10	1	110	/	/	~10	16
SS-V ₂ O ₃ @C/Zn	20	20	5000	/	80	~10	17

3D-Zn	5	5	160	/	/	/	18
Ti ₃ C ₂ T _x MXene/ZnS	10	1	180	/	/	~40	19
N-VG@CC	1	1	70		/	~10	20
FPCH-ZI/Zn	10	5	650	43.5	51	/	21
3D-RFGC@Zn	1	1	7300	15	20	~5	This work
3D-RFGC@Zn	20	20	2000	32	50	~10	This work
3D-RFGC@Zn	40	40	2200	26	100	~20	This work
3D-RFGC@Zn	80	80	2600	31	100	~40	This work

References

1. Tao Y, *et al.* Atomically Dispersed Cu in Zeolitic Imidazolate Framework Nanoflake Array for Dendrite-Free Zn Metal Anode. *Small* **18**, e2203231 (2022).
2. Sun PX, *et al.* Formation of Super-Assembled TiO(x)/Zn/N-Doped Carbon Inverse Opal Towards Dendrite-Free Zn Anodes. *Angew Chem Int Ed Engl* **61**, e202115649 (2022).
3. Zhang G, Zhang X, Liu H, Li J, Chen Y, Duan H. 3D-Printed Multi-Channel Metal Lattices Enabling Localized Electric-Field Redistribution for Dendrite-Free Aqueous Zn Ion Batteries. *Advanced Energy Materials* **11**, (2021).
4. Zhou J, *et al.* Encapsulation of Metallic Zn in a Hybrid MXene/Graphene Aerogel as a Stable Zn Anode for Foldable Zn-Ion Batteries. *Adv Mater* **34**, e2106897 (2022).
5. Xie F, *et al.* Mechanism for Zincophilic Sites on Zinc-Metal Anode Hosts in Aqueous Batteries. *Advanced Energy Materials* **11**, (2021).
6. Kang Z, *et al.* 3D Porous Copper Skeleton Supported Zinc Anode toward High Capacity and Long Cycle Life Zinc Ion Batteries. *ACS Sustainable Chemistry & Engineering* **7**, 3364-3371 (2019).
7. An Y, Tian Y, Xiong S, Feng J, Qian Y. Scalable and Controllable Synthesis of Interface-Engineered Nanoporous Host for Dendrite-Free and High Rate Zinc Metal Batteries. *ACS Nano* **15**, 11828-11842 (2021).
8. Buke W, *et al.* High Zinc Utilization Aqueous Zinc Ion Batteries Enabled by 3D Printed Graphene Arrays. *Energy Storage Materials*, (2022).
9. Yang J-L, Yang P, Yan W, Zhao J-W, Fan HJ. 3D zincophilic micro-scaffold enables stable Zn deposition. *Energy Storage Materials* **51**, 259-265 (2022).
10. Zeng Y, *et al.* Dendrite-Free Zinc Deposition Induced by Multifunctional CNT Frameworks for Stable Flexible Zn-Ion Batteries. *Adv Mater* **31**, e1903675 (2019).
11. Chen G, *et al.* Reversible and homogenous zinc deposition enabled by in-situ grown Cu particles on expanded graphite for dendrite-free and flexible zinc metal anodes. *Energy Storage Materials* **50**, 589-597 (2022).
12. Ling W, *et al.* An ultrahigh rate dendrite-free Zn metal deposition/stripping enabled by silver nanowire aerogel with optimal atomic affinity with Zn. *Energy Storage Materials* **51**, 453-464 (2022).
13. Yan M, *et al.* Constructing Three-Dimensional Topological Zn Deposition for Long-Life Aqueous Zn-Ion Batteries. *ACS Appl Mater Interfaces* **14**, 51010-51017 (2022).
14. Xue P, *et al.* A MOF-Derivative Decorated Hierarchical Porous Host Enabling Ultrahigh Rates and Superior Long-Term Cycling of Dendrite-Free Zn Metal Anodes. *Adv Mater* **34**, e2110047 (2022).
15. Zeng L, *et al.* Direct 3D printing of stress-released Zn powder anodes toward flexible dendrite-free Zn batteries. *Energy Storage Materials* **54**, 469-477 (2023).
16. Jian Q, Guo Z, Zhang L, Wu M, Zhao T. A hierarchical porous tin host for dendrite-free, highly reversible zinc anodes. *Chemical Engineering Journal* **425**, (2021).

-
17. Hong C, Yang G, Wang C. Highly Reversible Zn Electrodeposition Enabled by an Artificial 3D Defect-Rich Conductive Scaffold. *ACS Appl Mater Interfaces* **13**, 54088-54095 (2021).
 18. Bayaguud A, Luo X, Fu Y, Zhu C. Cationic Surfactant-Type Electrolyte Additive Enables Three-Dimensional Dendrite-Free Zinc Anode for Stable Zinc-Ion Batteries. *ACS Energy Letters* **5**, 3012-3020 (2020).
 19. An Y, Tian Y, Liu C, Xiong S, Feng J, Qian Y. Rational Design of Sulfur-Doped Three-Dimensional $Ti_{(3)}C_{(2)}T_{(x)}$ MXene/ZnS Heterostructure as Multifunctional Protective Layer for Dendrite-Free Zinc-Ion Batteries. *ACS Nano* **15**, 15259-15273 (2021).
 20. Cao Q, *et al.* Regulating Dendrite-Free Zinc Deposition by 3D Zincophilic Nitrogen-Doped Vertical Graphene for High-Performance Flexible Zn-Ion Batteries. *Advanced Functional Materials* **31**, (2021).
 21. Park JB, Choi C, Park JH, Yu S, Kim DW. Synergistic Design of Multifunctional Interfacial Zn Host toward Practical Zn Metal Batteries. *Advanced Energy Materials* **12**, (2022).

Responses to the comments by Reviewer #2

Reviewer #2: The authors answered to my main concerns. Nonetheless, still some points remain to be improved:

1) This is fine

2) The color doesn't match between the caption and data of Figure 12a. In addition, the main concern is that the scale of y-axis is different in Figure 12b-e. Please match the y-axis scale to be same (maximum voltage to be 0.1 and minimum to be -0.1). Furthermore, I still doubt that the performance of your data does really exceed the other recently published papers in Nature Communications.

3) This is fine

4) Same as review #2, I recommend you to match the scale of y-axis of Figure 14a-d, and it's better to use “,” than “~” for areal capacity. I wonder in figure 14a, the step time seems to be changed before and after 10 hours. I think you have to support about the difference. Also, as I understand your answer, the cell short-circuit occurred only in figure 14d, you have to mention about your criteria of cell short-circuit. At last, you need to explain more about the voltage curve fluctuation during the cycle, because it occurs at every current density in figure 14a-d.

5) This is fine

6) This is fine

7) This is fine

Due to the comments above, I suggest still a major revision, in order to fulfill the aforementioned points of 2 and 4.

Response # (2): Thank you for your further comments. We have matched the color between the caption and data of Figure 12a, and the y-axis scale has been revised to be the same from -0.1 V to 0.1 V (**Figure 1**). The electrochemical performance in our manuscript, in fact, is highly competitive compared to other reported 3D current collectors for zinc anodes in terms of coulombic efficiency and cycling stability in half-cells, rate capability, current density, areal capacity, and long-term lifespan in symmetric cells, excellent suitability in full-cells (**Figures 3I and 4I in the revised manuscript**). In particular, we compare recent work on zinc anode on *Nature Communications* (e. g., *Nat Commun.* 14, 641 (2023); *Nat Commun.* 14, 76 (2023); *Nat Commun.* 13, 7922 (2022); *Nat Commun.* 13, 5348 (2022); *Nat Commun.* 13, 3252 (2022).) (**Table 1**), and our proposed graphene-modified 3D matrices possess good structural originality and excellent electrochemical properties, especially in the high deposition-stripping current density and capacity, this is far beyond the current reported researches of zinc anodes. The performance of the 1500 cycles at 10 A g⁻¹ in the V₂O₅@3D-LC/3D-RFGC@Zn full cell and the 20000 cycles in AC@3D-LC/3D-RFGC@Zn

capacitor show great promise for high-performance zinc metal anodes.

Figure 1. (a) The voltage curves of bare Zn, 3D Zn foam, 3D-LFGC@Zn and 3D-RFGC@Zn anodes and (b-e) enlarged voltage curves at different stages in symmetric cells.

Response # (4): We have matched the scale of y-axis to be the same from -1.0 V to 1.0 V (**Figure 2**). We are also revised “~” to “,” for areal capacity in **Figures 2a-2d**. The difference of step times you find in Figure 4a does exist, which is because that we set the cell test step to a “rate mode”, as shown in the graph we have marked, where the current was changed to 2 mA cm^{-2} from the fifth cycle. For the short-circuit in 3D Zn foam symmetric cells, in the last revised manuscript, we described what we really meant was that the short-circuit occurred under the four working conditions of **Figures 2a-2d**. When the current density was 1 mA cm^{-2} and the capacity was 1 mAh cm^{-2} , we observed that a uniform zinc deposition and stripping process took place in 3D Zn foam symmetric cells at the beginning of the cycle, but with the reaction went on, there was

already a micro-short circuit at 5th cycle, and then a sudden voltage spike when the current changed to 2 mA cm⁻² (6th cycle), followed by a large current fluctuation shortly thereafter, which was indicative of a short-circuit (**Figure 2a**). Furthermore, when we increased the current to 10 mA cm⁻², 20 mA cm⁻² and 80 mA cm⁻², the rapid voltage fluctuation indicated that the deposition and stripping process of zinc metal at these currents and capacities was blocked and could not be completed uniformly and stably. This was also demonstrated by SEM images of 3D zinc foams (**Figure 3**). The zinc foam showed a very rough surface, which is not conducive to homogenizing current distributions. Especially, when the working current was large, the local current density in the protuberance was too high, which led to the nucleation and growth of zinc dendrite, and subsequently the whole deposition process being blocked, thus resulting in huge voltage fluctuation (**Figures 2b-2d**). As for the criteria of cell short-circuit, a short-circuit occurred in symmetric cells can be determined by large voltage fluctuation without recovery, according to previous reports in ZMAs (*Joule* 2022, 6, 269-279; *Angew. Chem. Int. Ed.* 2022, 61, e202202780.).

Figure 2. Symmetrical cells of 3D Zn foam operating at a) 1 mA cm⁻² and 1 mAh cm⁻²; b) 10 mA cm⁻² and 10 mAh cm⁻²; c) 20 mA cm⁻² and 20 mAh cm⁻²; d) 80 mA cm⁻² and 80 mAh cm⁻².

Figure 3. SEM images of commercial 3D foam Zn from the top-view and cross-view.

Table 1. Performance comparison of our 3D-RFGC@Zn anode and recently reported ZMAs in *Nature Communications* (Cd: current density; C: capacity; L: lifespan; C: cycles in half cells, symmetric cells and full cells)

Samples	Cd (mA cm ⁻²)	C (mAh cm ⁻²)	L (h)	cathode	Current density (A g ⁻¹ /C)	Cycles (n)	Ref.
PVDF-Sn@Zn	1	1	1200	MnO ₂ @C	2	700	1
	5	5	500	/	/	/	
	10	10	200	/	/	/	
	5	5	700	/	/	/	
Sb/Sb ₂ Zn ₃ @Cu	20	10	550	Br ₂	10	800	2
	200	10	5(cycles)	Br ₂	20	170	
	50	50	220	/	/	/	
3D Zn-Cu	1	/	1000	MnO ₂	5C	600	3
	/	/	/	MnO ₂	10C	400	
	/	/	/	MnO ₂	20C	1500	
	1	1	1200	V ₂ O ₅	1C	1000	
C ₃ N ₄ QDs	/	/	/	MnO ₂	1C	500	4
	/	/	/	VOPO ₄	1	3000	
La(NO ₃) ₃ additive	1	1	1200	VS ₂	1	1500	5
	10	5.93	160	/	/	/	
	1	0.5	400	/	/	/	
	10	0.5	1000	/	/	/	
3D-RFGC@Zn (This work)	40	0.5	3000	/	/	/	Half-cell
	40	8	50	/	/	/	
	40	40	20	/	/	/	
	80	0.5	5000	/	/	/	
	120	0.5	2000	/	/	/	
	120	5	150	/	/	/	
	1	1	7300	/	/	/	Symmetric cells
	20	20	2000	/	/	/	

	40	40	2200	/	/	/	
	80	80	1600	/	/	/	
	/	/	/	V ₂ O ₅	40 mA cm ⁻²	2400	
	/	/	/	MnO ₂	4 mA cm ⁻²	500	Full cells
	/	/	/	AC (capacitor)	40 mA cm ⁻²	20000	

References

1. Cao Q, *et al.* Gradient design of imprinted anode for stable Zn-ion batteries. *Nat Commun* **14**, 641 (2023).
2. Zheng X, *et al.* Constructing robust heterostructured interface for anode-free zinc batteries with ultrahigh capacities. *Nat Commun* **14**, 76 (2023).
3. Tian H, *et al.* Three-dimensional Zn-based alloys for dendrite-free aqueous Zn battery in dual-cation electrolytes. *Nat Commun* **13**, 7922 (2022).
4. Zhang W, *et al.* Self-repairing interphase reconstructed in each cycle for highly reversible aqueous zinc batteries. *Nat Commun* **13**, 5348 (2022).
5. Zhao R, *et al.* Lanthanum nitrate as aqueous electrolyte additive for favourable zinc metal electrodeposition. *Nat Commun* **13**, 3252 (2022).

Responses to the comments by Reviewer #3

Reviewer #3: The reviewer appreciates the authors' efforts and is satisfied with most responses and revisions. I do have a few more minor comments for the authors to consider. After adequately addressing these comments, the manuscript should be suitable for publication.

1. Fig. 3l needs to be simplified and clarified. For example, it took me a while to find the correct y-axis to read the values of the columns or data points.
2. In Fig. 4l, the authors use the size of circles to represent the values of areal capacities, which needs to be clarified. I suggest putting a scale bar so that the readers can measure the diameters of the circles, or change to other clearer representations. I also recommend putting all the data of Fig. 4l in a table in the supporting information.
3. Looking at the schematic illustration of Fig. 7a, the readers might think the authors use the 3D-LFGC@Zn anode, while in fact, the electrode is 3D-RFGC@Zn. Please correct this misleading figure.

Response (1): Thank you for your further comments. In the original version, we combined the current density, areal capacity, number of cycles, and average Coulomb efficiency into one graph, and it was really hard to tell what each meant quickly and clearly, which is extremely inconvenient for the readers. **Figure 3l** (revised manuscript) has been simplified to compare the performance of 3D-RFGC with previously reported literature in terms of the current density and cycle capability of these two indicators, as shown below (**Figure 1**). The corresponding data and the data that does not appear in this figure are summarized in **Table S2** in the revised supporting information.

Figure 1. The comparison of current densities of deposition, and cycle capability for 3D-RFGC with previously reported literature.

Response (2): We have added a scale bar in the figure so that the readers can measure the diameters of the circles (**Figure 2**), and all the data of **Figure 4l** have summarized

as Table S5 in the revised supporting information.

Figure 2. The comparison of current densities, areal capacities and cycle capability in symmetrical cells with previously reported 3D Zn anodes literature (The size of the circles represents the values of areal capacities).

Response (3): We have corrected the error made in Figure 7a, as shown below (**Figure 3**). In the schematic illustration of full cell in this manuscript, we used 3D-RFGC@Zn as anode, and V₂O₅ (MnO₂, AC)@3D-LC as cathode.

Figure 3. The structural design of full cells using 3D matrices for both anode and cathode.

We want to thank the reviewers for the valuable comments and suggestions, which are helpful for improving the quality of this manuscript.

REVIEWERS' COMMENTS

Reviewer #2's comment:

The authors have addressed the related concerns and the manuscript improved a lot, it is recommended for publication.